Corrected: Publisher correction

# *Leishmania* RNA virus exacerbates Leishmaniasis by subverting innate immunity via TLR3-mediated NLRP3 inflammasome inhibition

Renan V.H. de Carvalho [1], Djalma S. Lima-Junior [1], Marcus Vinícius G. da Silva[1], Marisa Dilucca [1], Tamara S. Rodrigues[1], Catarina V. Horta [1], Alexandre L.N. Silva[1], Patrick F. da Silva [2], Fabiani G. Frantz [2], Lucas B. Lorenzon [1], Marcos Michel Souza [1], Fausto Almeida[3], Lilian M. Cantanhêde [4], Ricardo de Godoi M. Ferreira [4], Angela K. Cruz[1] & Dario S. Zamboni [1]*

*Leishmania* RNA virus (LRV) is an important virulence factor associated with the development of mucocutaneous Leishmaniasis, a severe form of the disease. LRV-mediated disease exacerbation relies on TLR3 activation, but downstream mechanisms remain largely unexplored. Here, we combine human and mouse data to demonstrate that LRV triggers TLR3 and TRIF to induce type I IFN production, which induces autophagy. This process results in ATG5-mediated degradation of NLRP3 and ASC, thereby limiting NLRP3 inflammasome activation in macrophages. Consistent with the known restricting role of NLRP3 for *Leishmania* replication, the signaling pathway triggered by LRV results in increased parasite survival and disease progression. In support of this data, we find that lesions in patients infected with LRV+ *Leishmania* are associated with reduced inflammasome activation and the development of mucocutaneous disease. Our findings reveal the mechanisms triggered by LRV that contribute to the development of the debilitating mucocutaneous form of Leishmaniasis.

[1] Departamento de Biologia Celular e Molecular e Bioagentes Patogênicos, Faculdade de Medicina de Ribeirão Preto, Universidade de São Paulo, Ribeirão Preto, São Paulo, Brazil. [2] Laboratório de Imunologia e Epigenética, Departamento de Análises Clínicas, Toxicológicas e Bromatologia, Faculdade de Ciências Farmacêuticas de Ribeirão Preto, Universidade de São Paulo, Ribeirão Preto, SP, Brazil. [3] Departamento de Bioquímica e Imunologia, Faculdade de Medicina de Ribeirão Preto, Universidade de São Paulo, Ribeirão Preto, São Paulo, Brazil. [4] Fundação Oswaldo Cruz, Unidade Rondônia, Porto Velho, Rondônia, Brazil. *email: dszamboni@fmrp.usp.br

Leishmaniasis is a disease endemic in 98 different countries, causing ~70,000 deaths per year[1,2]. It is caused by parasites from *Leishmania* genus, being transmitted by the bite of phlebotomine sand flies vectors[3]. Once injected in the skin dermis, parasites are rapidly internalized by macrophages, and may establish infection if they resist to the hostile endosomal environment where they are found[4]. The clinical manifestations range from self-healing cutaneous lesions to more aggressive forms of the disease, such as mucocutaneous lesions and visceral pathology, eventually causing death[5]. The outcome of the disease varies according to the infective specie and intrinsic virulence factors, as well as the immune response displayed by the infected mammalian host[6,7].

The innate immune system plays an essential role in controlling parasitic infections[4,8–11], since it is equipped with a wide variety of pattern-recognition receptors (PRRs), such as Toll-like-receptors (TLRs) and Nod-like receptors (NLRs), that can rapidly induce cytokine release, inflammation and parasite control[12–14]. TLRs are able to recognize several pathogen-associated molecular patterns (PAMPs) and are found in the cellular surface and endosomal membranes, inducing the production of inflammatory mediators (pro-IL-1β, TNF-α, IL-6, chemokines) through NF-κB, which contributes to inflammation by itself and by working as a first signal to induce NLR's activation[15]. The majority of TLRs signals exclusively via MyD88, except for TLR3, which recognizes double-stranded RNA (dsRNA) and recruits TRIF, triggering a strong antiviral response via IRFs and type I IFN[16]. Recognition of *Leishmania* by TLR2, 3, 4, 7, and 9 is described as important effector mechanisms to restrict parasite control[17,18]. On the other hand, IFN-α and IFN-β, which play an essential role in the control of virus and some bacteria, are detrimental during *Leishmania* infection, yet the mechanisms responsible for this phenotype are still poorly understood[19,20].

Among the NLR family, NLRP3 is the best characterized, being involved in a wide variety of pathological conditions, such as Alzheimer's disease, cancer, autoimmune disorders, and infectious diseases[21–24]. Several groups report that the NLRP3 inflammasome is activated by *Leishmania spp.* and plays a crucial role in the outcome of the infection[25–29]. Besides, autophagy, which is a cellular homeostatic process induced under starvation conditions and infections[30], is triggered by *Leishmania spp.*, but its role during this parasitic infection is still controversial. Although some studies suggest that it plays a crucial role in parasite control[18], others claim that *Leishmania* uses autophagic machinery to improve its replication[31–33]. Of note, it is well established in the literature that autophagy plays an important role in modulating inflammasome activation[34], but whether *Leishmania*-induced autophagy regulates inflammasome activation is still unknown.

Patients infected with *Leishmania* (*Viannia*) *braziliensis* and *Leishmania* (*Viannia*) *guyanensis* may display cutaneous lesions that can disseminate and reach nasopharyngeous tissues, causing mucocutaneous Leishmaniasis, a disfiguring form of the disease that is much more difficult to treat[3]. Almost three decades ago, it was discovered that some strains of *L. guyanensis* harbor an endosymbiontic double-stranded RNA virus that belongs to *Totiviridae* family, being termed *Leishmania* RNA virus (LRV)[35]. LRV relevance in the course of Leishmaniasis is still poorly known, but some recent studies in murine models show that LRV presence is intimately related to the severity of the infection, altering the course of the host immune response to the parasite[36–39]. Although TLR3 has a minor role in detecting and responding to *Leishmania spp.* compared with TLR9, upon *L. guyanensis* LRV+ infection, TLR3 senses dsRNA from LRV in the endosomal compartment, triggering a robust inflammatory response, with the production of TNF-α and type I IFN,

exacerbating disease in mice[36,40]. However, the mechanisms by which LRV promotes parasite persistence and increased disease remain largely unexplored[37,41].

Using a mutation-free LRV positive clone from the *L. guyanensis* strain M4147 (*L.g.*+), we randomly generate a clone lacking LRV (*L.g.*−), allowing us to study the impact of LRV presence in the outcome of *Leishmania* infection. We determine the innate immune signaling pathways and mechanisms triggered by LRV that allows *L.g.*+ to promote disease in a murine model of Leishmaniasis.

## Results

**The inflammasome associates with disease severity in humans.** To investigate the participation of the inflammasome in the severity of the disease, we obtained clinical material from 49 patients from the state of Rondônia, Brazil (Supplementary Table 1), and measured the levels of active caspase-1 (Casp1) and IL-1β in the lesion aspirates from patients with the cutaneous leishmaniasis (CL) and mucocutaneous leishmaniasis (MCL). Strikingly, lesions of CL patients contained higher levels of IL-1β compared with MCL lesions, but no differences between these two groups were observed in active Casp1 (Fig. 1a, b). Sixty seven percent of the MCL patients were infected with LRV+ parasites and 33% of the MCL patients were infected with LRV− parasites (Fig. 1c and Supplementary Table 1). Patients infected with LRV+ parasites displayed lower levels of IL-1β and active Casp1 as compared with patients infected with LRV− parasites (Fig. 1d, e). The levels of TNF-α, an inflammasome-independent cytokine, did not differ between LRV+ and LRV− patients (Fig. 1f). Importantly, only 10% of these patients infected with LRV− parasites developed mucocutaneous lesions, in contrast to 31% of the patients infected with LRV+ (Fig. 1g). Collectively, these data obtained with clinical samples indicates that inflammasome activation is directly correlated with disease severity, and the presence of the LRV influences inflammasome activation and disease development. These data prompted us to further investigate the mechanisms underlying these processes.

**LRV promotes disease severity and parasite survival.** NLRP3 activation has been extensively investigated using murine bone marrow-derived macrophages (BMDMs) and a mouse model of Leishmaniasis[25–27,42]. The reference strain of *L. guyanensis* M4147 is known to harbor high levels of a *Leishmania* RNA virus (LRV), which has been previously found to influence disease severity[36,37]. Therefore, we generated a spontaneous clonal strain of *L.g.* M4147 that does not contain LRV (Supplementary Fig. 1a). We generated a clone (clone 40) that is negative for the virus (*L.g.*−) (Supplementary Fig. 1b). We confirmed the PCR data using immunofluorescence, since high amounts of dsRNA in the parental strain, but not in clone 40, were observed (Supplementary Fig 1c, d). Interestingly, loss of LRV did not affect *L.g.* capacity to proliferate in axenic culture (Supplementary Fig. 1e).

The generation of a virus-free clone from M4147 WT strain provided us with a powerful tool to assess the importance of LRV in the course of *L.g.* infection. Thus, we infected C57BL/6 mice with stationary-phase promastigotes of M4147 WT (*L.g.*+) and clone 40 (*L.g.*−), and assessed lesion development for 4 weeks. We found that mice infected with *L.g.*+ display increased development of lesions in the ears compared with *L.g.*− (Fig. 2a). Accordingly, parasite loads in the ear (Fig. 2b) and draining lymph node (Fig. 2c) were higher in mice infected with *L.g.*+. Similar results were obtained when mice were infected with metacyclic promastigotes (Fig. 2d–f). BMDMs were infected with metacyclic parasites of *L.g.*− or *L.g.*+ and the infection was monitored for 96 h. We found that LRV does not affect

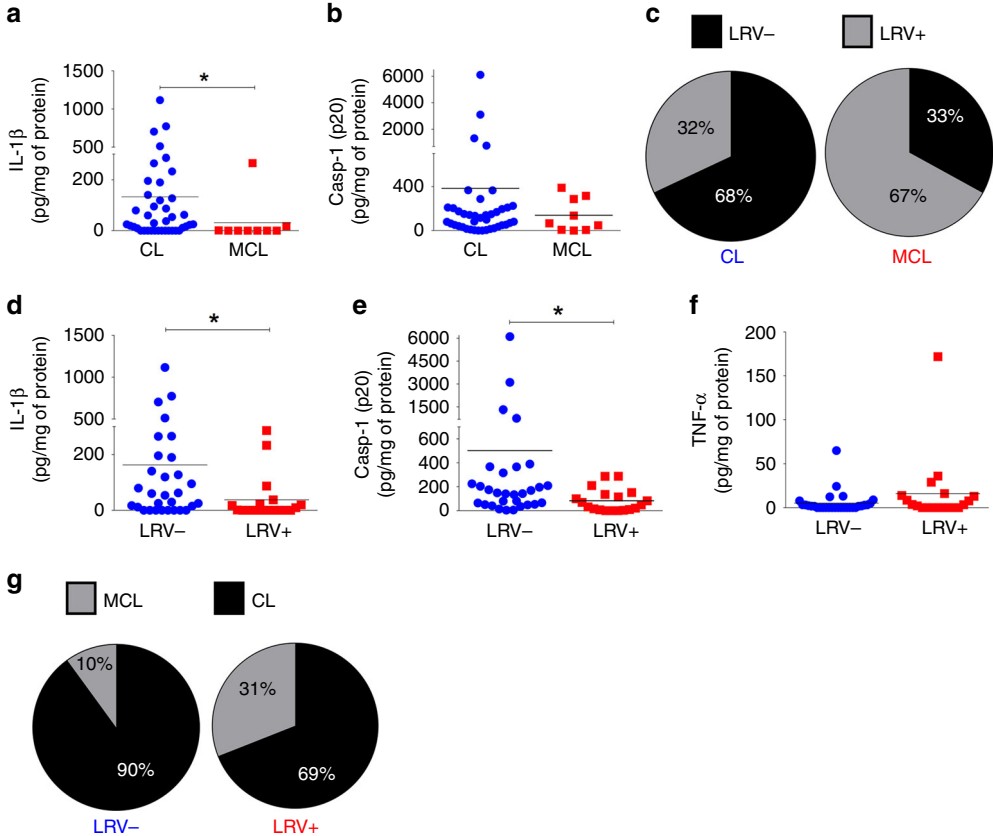

**Fig. 1** LRV and MCL associate with decreased levels of IL-1β and Casp1 p20. **a–c** Patients were grouped according to the outcome of the disease (CL = Cutaneous Leishmaniasis; MCL = Mucocutaneous Leishmaniasis), regardless of the infecting *Leishmania* specie. Levels of IL-1β (**a**) and cleaved Casp1 (**b**) were quantified by ELISA in aspirates collected from lesions from both CL ($n = 40$) and MCL ($n = 9$) patients, and the frequency of LRV+ and LRV− parasites among each group is shown (**c**). **d–g** The same group of patients was distributed according to their LRV status, regardless of the infecting *Leishmania* specie. Levels of IL-1β (**d**), cleaved Casp1 (**e**), and TNF-α (**f**) were plotted from both LRV+ ($n = 19$) and LRV− ($n = 30$) patients, and the frequency of CL and MCL among each group is shown (**g**). Each value was corrected to the respective amount of protein (determined by Bradford) from each patient. Statistical analysis was performed by Mann–Whitney's test. $P < 0.05$ (*) was considered statistically significant

*L.g.* internalization (1 h) (Fig. 2g, h). However, *L.g.*+ is more resistant to BMDMs killing from 24 to 96 h of infection (Fig. 2g). Accordingly, the average numbers of parasites per cell is higher in macrophages infected with *L.g.*+ at later times of infection (Fig. 2h). We then verified whether these findings were due to better macrophage survival upon *L.g.*+ infection. FACS analysis of non-infected and infected cells stained with the Live/Dead viability dye and CFSE show that while we observe very little cell death after 48 h of infection, LRV does not impact this process (Supplementary Fig. 2) (Fig. 2i–m). Taken together, these results suggest that LRV promotes pathogenesis and accounts for parasite resistance in macrophages and in vivo.

**LRV dampens inflammasome activation by *L. guyanensis*.** Next, we assessed inflammasome activation in response to infection by *L.g.*+ and *L.g.*−. We found that infection with *L.g.*− induced a more robust IL-1β production when compared with *L.g.*+ (Fig. 3a), an effect observed as early as 9 h after infection (Fig. 3b). Importantly, inflammasome activation can only be robustly achieved upon previous priming (Supplementary Fig. 3a), just as observed in others studies[27,43,44]. The presence of the virus did not affect the production of inflammasome-independent cytokines, such as TNF-α (Fig. 3c). The reduced production of IL-1β in response to *L.g.*+ was also observed when macrophages were infected with different MOI of metacyclic promastigotes (MC) (Fig. 3d–f), using different MOIs of stationary-phase parasites (Supplementary Fig. 3b), when macrophages were primed with

different stimuli (Supplementary Fig. 3c) and finally when we used macrophages from different mouse strains (Supplementary Fig. 3d).

To evaluate Casp1 activation directly, we stained the infected macrophages with FAM-YVAD[45]. *L.g.*+ induced less Casp1 activation in BMDMs when compared with *L.g.*− (Fig. 3g–i), We also assessed the cleavage of Casp1 and IL-1β by western blot, and found that LRV presence impairs both (Fig. 3j). Next, we evaluated whether LRV effects in inflammasome activation affects IL-1β production induced by a different microorganism. We co-infected BMDMs with *L.g.*+ or *L.g.*− plus *flaA⁻ Legionella pneumophila*, a gram-negative bacteria known to trigger NLRP3[46,47]. Co-infection of *L.g.*− with *flaA⁻ L. pneumophila* promoted an additive effect on IL-1β secretion, however, the co-infection of *L.g.*+ with *flaA⁻ L. pneumophila* significantly decreased inflammasome activation when compared with cells infected with *flaA⁻ L. pneumophila* alone (Fig. 3k), To evaluate if clonal variations, rather than LRV presence, contribute to inhibition of inflammasome activation, we generated five clones of M4147 harboring LRV (Fig. 3l). We infected BMDMs with the 5 clones and measured IL-1β secretion upon infection. We found that *L.g.*+ and all LRV + M4147-derived clones induce less inflammasome activation than *L.g.*− (Fig. 3m), suggesting that LRV presence, rather than clonal characteristics of the strain, affect inflammasome activation.

To investigate whether the expression of pro-IL-1β is affected by LRV, we infected BMDMs for 6 or 24 h with *L.g.*+ or

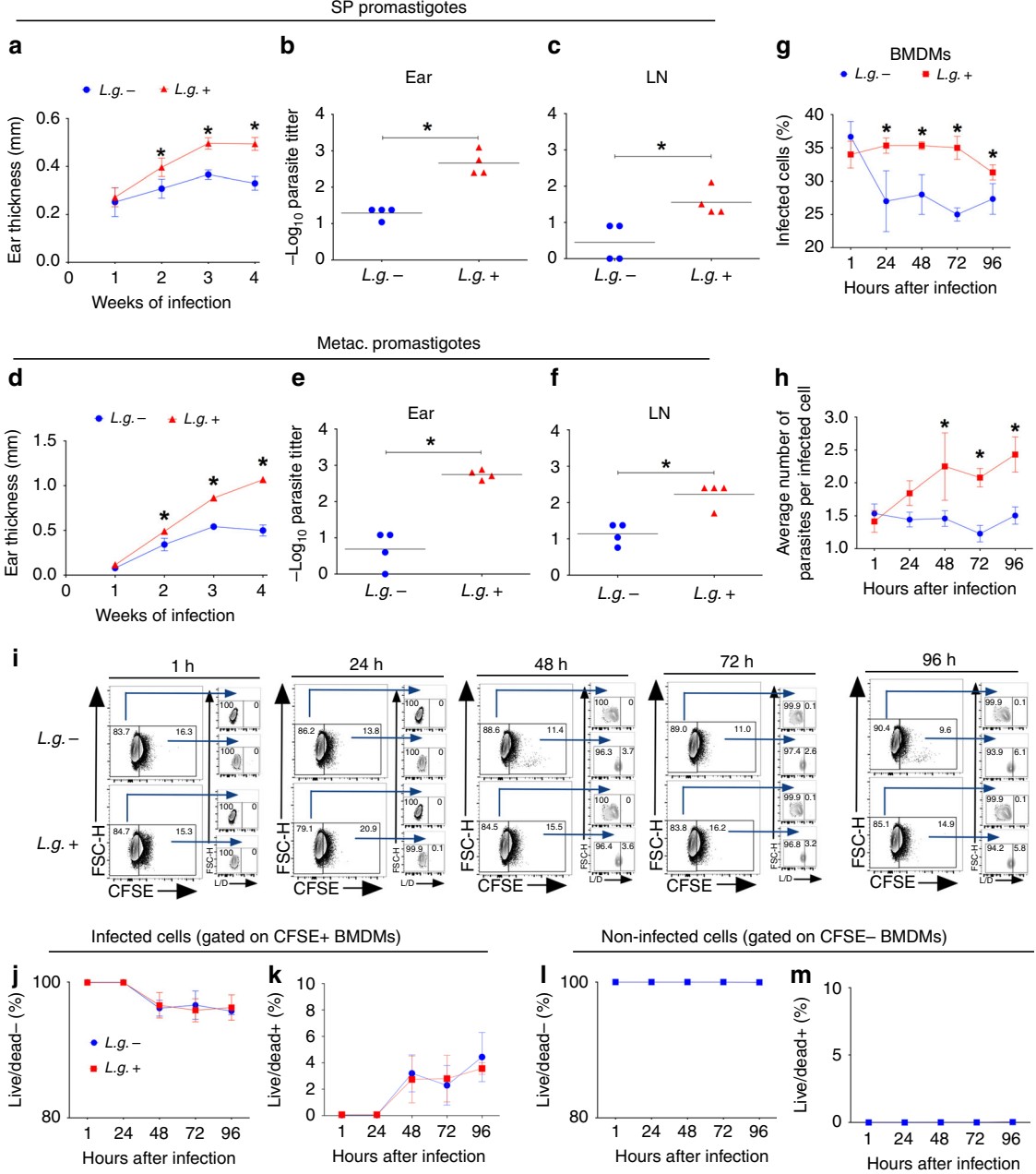

**Fig. 2** LRV increases *L.g.* virulence in vitro and in vivo. **a–f** C57BL/6 mice (*n* = 4 per group) were injected intradermally in the left ear with SP (**a–c**) or metacyclic promastigotes (**d–f**) of *L.g.−* or *L.g.+* and ear thicknesses were followed weekly (**a**, **d**). At 4 w.p.i., parasite titers were determined in the infected ear (**b**, **e**) and draining lymph node (**c**, **f**). **g–m** WT BMDMs were infected with metacyclic promastigotes of *L.g.−* and *L.g.+* at a MOI of 1. Cells were washed after 1 h of infection, and kept in culture for 1, 24, 48, 72, or 96 h. **g, h** In vitro killing assay (Panotico *Giemsa*) showing the percentage of infected cells (**g**) and the average number of parasites per cell (**h**) in the end of each time point are shown. (**i–m**) FACS analysis of BMDMs infected with *L.g.−* and *L.g.+* that were pre-incubated with CFSE dye prior to infection. The representative contour plot **i** and the percentages of alive and dead infected (**j**, **k**) or non-infected (**l**, **m**) cells are shown. One representative of three (**a–f**) or two (**g–m**) independent experiments performed is shown, with biological (**a–f**) or technical (**g–m**) replicates. The results are shown as mean ± SD. Statistical analysis was performed by two-way ANOVA with Bonferroni's multiple comparison test. *P* < 0.05 was considered statistically significant (*)

*L.g.−* and measured the transcription of *Il1b* by qRT-PCR. Interestingly, LRV accounts to increased *Il1b* expression at early times of infection (Supplementary Fig. 4a, b). In addition, *L.g.+* induced higher expression of *Tnfa*, *Il12*, and *Ifnb* (Supplementary Fig. 4c-h). These data suggest that LRV-mediated inhibition of inflammasome activation does not occur at a transcriptional level. The production of ROS and the potassium efflux are known to be important second signals for NLRP3 activation by *Leishmania* spp.[25,46,47]. Thus, we infected BMDMs with *L.g.−* or *L.g.+* and

measured total ROS and mitochondrial ROS production. Both clones induce similar production of total ROS, as shown by the representative histogram and the iMFI (Supplementary Fig. 5a, b). Interestingly, none of the clones induced mitochondrial ROS (Supplementary Fig. 5c). Using asante potassium green-2 dye, which specifically stains intracellular $K^+$[48], we found that both *L.g.−* and *L.g.+* trigger a similar efflux of $K^+$ in the infected cells, similar to the pore-forming toxin nigericin (Supplementary Fig. 5d). These data suggest that the LRV-mediated inhibition

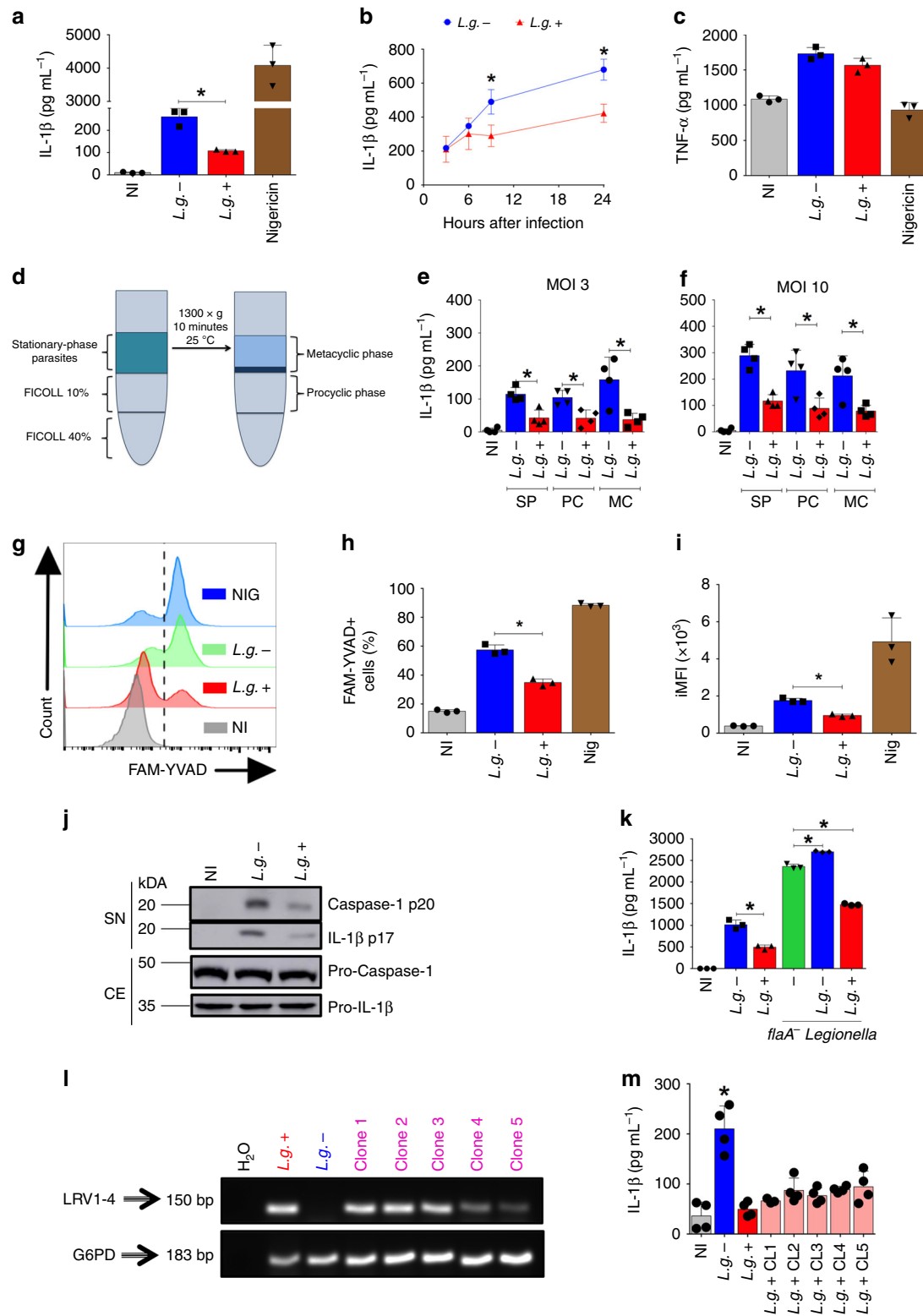

of inflammasome activation occur neither via the first signal nor via the second signal.

**TLR3 mediates LRV-induced inhibition of inflammasome activation.** It was previously demonstrated that LRV is sensed by TLR3 upon *L.g.* infection[36]. Thus, we tested if TLR3 is required for LRV-mediated inhibition of inflammasome activation. We infected BMDMs from C57BL/6 and *Tlr3*[−/−] mice and assessed

inflammasome activation. Macrophages from *Nlrp3*[−/−] mice were used as control. We found that the reduced inflammasome activation in response to *L.g.+* was abolished in the absence of *Tlr3*[−/−]. This was evident when we measured Casp1 activation by FAM-YVAD (Fig. 4a, b) and WB (Fig. 4c). Additionally, we infected these cells with stationary-phase (SP) or metacyclic (Metac.) parasites, and found similar results for IL-1β release (Fig. 4d, e). We used the co-infection with *flaA*[−] *L. pneumophila*

**Fig. 3** LRV limits inflammasome activation by *L.g.* in BMDMs. **a**, **b** C57BL/6 BMDMs were primed for 4 h with LPS (500 ng/mL) and infected with Stationary-phase parasites (SP) of *L.g.*− or *L.g.*+ at MOI 10 for 24 h (**a**) or 3, 9 and 24 h (**b**). Nigericin (20 μM) was used as a positive control. Cell-free supernatants were collected and ELISA for IL-1β (**a**, **b**) and TNF-α (**c**) was performed. **d**–**f** Metacyclic promastigotes were purified by FICOLL gradient (**d**), and IL-1β levels were quantified in BMDMs infected by either Stationary-phase (SP), Procyclic (PC) or Metacyclic (MC) parasites at MOI 3 (**e**) or 10 (**f**). **g**–**i** FLICA assay showing Casp1 activation upon infection with *L.g.*− and *L.g.*+. The representative histogram (**g**), the percentage of Casp1 positive cells (FAM-YVAD+) (**h**) and integrated MFI (iMFI) (**i**) are shown. Nigericin (20 μM) was used as a positive control. **j** LPS-primed BMDMs were infected for 24 h with *L.g.* clones. Western blotting for cleaved Casp1 (p20) and IL-1β (p17) was performed in the cell-free SN, while pro-caspase-1 and pro-IL-1β were measured in the cellular extracts (CE). **k** BMDMs were primed and infected with *L.g.*− or *L.g.*+ at a MOI of 10. After 20 h of infection, cells were infected with *L. pneumophila* (*flaA*− *Legionella*). 4 h later, cell-free supernatants were collected and the levels of IL-1β were assessed by ELISA. **l**, **m** Mouse BMDMs were left non-infected (NI) or infected with *L.g.*−, *L.g.*+ or five different LRV+ clonal strains. **l** LRV and G6PD (*Leishmania*) amplification product is shown by PCR. Water (H₂O) was used as a negative control. **m** IL-1β levels were measured by ELISA. The results are shown as mean ± SD. Statistical analysis was performed by unpaired Student's *t* test, and *P* < 0.05 (*) was considered statistically significant. One representative of at least three independent experiments performed in technical triplicates or quadruplicates is shown

assay and confirmed that TLR3 is required for inhibition of inflammasome activation by *L.g.*+ (Fig. 4f). Furthermore, we found that addition of Poly:IC, a synthetic TLR3 ligand, complemented the defects of *L.g.*− in inflammasome inhibition (Fig. 4g). TLR3 activation is known to signal via TRIF to induce type I interferon production[49]. Therefore, we tested if IFN-β is sufficient to complement *L.g.*− for inhibition of inflammasome activation. Similar to Poly:IC, we found that IFN-β treatment restored the inhibition of inflammasome activation in response to *L.g.*− (Fig. 4h).

Next, we tested if the increased replication of *L.g.*+ in macrophages required the NLRP3 inflammasome and the TLR3-TRIF-IFN-β pathway. We infected BMDMs from C57BL/6, *Nlrp3*−/−, *Tlr3*−/− and *Trif*−/− mice with *L.g.*− or *L.g.*+ and measured intracellular replication. We found that these molecules did not influence the internalization of the parasites at 1 h after infection (Fig. 4i, j), but we detected an increased replication of *L.g.*− in *Nlrp3*−/− cells at 48 h after infection (Fig. 4k, l). Importantly, the increased replication of *L.g.*+ compared with *L.g.*− was only detected in C57BL/6 macrophages, indicating that LRV-mediated pathogenesis requires TLR3, TRIF and NLRP3 (Fig. 4k, l). We also tested the effect of exogenous IFN-β and Poly:IC in parasite replication and found that both promote intracellular replication of *L.g.*− in BMDMs, but did not affect intracellular replication of *L.g.*+ (Supplementary Fig. 6a, b).

**LRV promotes degradation of NLRP3 and ASC via autophagy.** Several viruses have been shown to induce autophagy[30,34]. In addition, type I IFN is also known to trigger autophagy[50]. Thus, we investigated whether LRV promotes autophagy in macrophages infected by *L.g.*+. First, we transduced BMDMs with retrovirus encoding the autophagosomal marker LC3 fused with the green fluorescent protein (LC3-GFP) and used these cells to assess the induction LC3 puncta in response to infection, as described[18]. We found that *L.g.*− and *L.g.*+ promote LC3 puncta formation at 3, 9, and 24 h of infection and autophagy induction is significantly increased in *L.g.*+ infected cells compared with *L.g.*− at 9 and 24 h of infection (Fig. 5a, b). Interestingly, non-infected cells also have an increased number of LC3 puncta at 24 h after infection, which suggests that soluble mediators released upon *L.g.*+ infection, such as type I IFN, could also induce autophagy in bystander cells (Fig. 5a). LC3 is expressed as a 18 KDa protein (LC3-I) in the cytosol and during autophagic process it is converted to a 16 KDa protein (LC3-II)[30]. Thus, we performed western blot to evaluate induction of autophagy by *L.g.* clones and confirmed that *L.g.*+ is more efficient to induce autophagy (Fig. 5c). Rapamycin was used as positive control. We next treated the cultures with Poly:IC and IFN-β and found that stimulation of BMDMs with Poly:IC and IFN-β was sufficient to trigger autophagy and rescue the reduced autophagy induction

observed in *L.g.*− (Fig. 5d). Accordingly, TLR3 was required for LRV-mediated autophagy induction (Fig. 5e). Collectively, these data indicate that LRV promotes autophagy during *L. guyanensis* infection via TLR3/IFN-β signaling.

Autophagosomes can target cytosolic components to degradation, such as receptors and damaged or senescent organelles, being a key process regulating inflammasome assembly and activation[30,51]. Since *L.g.*+ shows increased autophagy and reduced inflammasome activation compared with *L.g.*−, we hypothesized that autophagy negatively regulates the levels of inflammasome components. We tested it by western blot and found that *L.g.*+ promoted an increased degradation of NLRP3 and ASC, but not Casp1, compared with *L.g.*− (Fig. 5f). This process was not detected in *Tlr3*−/− BMDMs (Fig. 5f). Importantly, increased degradation of inflammasome components by *L.g.*+ does not happen at a transcriptional level (Supplementary Fig. 7a–f).

To further test the effect of autophagy in the LRV-mediated inhibition of inflammasome activation, we silenced ATG5 in primary BMDMs and assessed inflammasome activation in response to *L.g.*+ and *L.g.*− infection. We used a lentiviral vector encoding a *shAtg5* sequence that efficiently silences ATG5 (Fig. 6a), and found that ATG5 silencing increased inflammasome activation in response to *L.g.* as determined by the production of IL-1β (Fig. 6b), confirming our assertion that autophagy negatively regulates inflammasome activation. Importantly, the reduced inflammasome activation observed in cells infected in *L.g.*+ compared with *L.g.*− was abolished in ATG5-silenced BMDMs (Fig. 6b).

Next, we used BMDMs from a mouse genetically depleted for *Atg5* in which *Atg5* floxed alleles were specifically deleted in myeloid cells after crossing with *LysM*Cre mice. These mice were *LysM*Cre/+/*Atg5*FL/FL, herein called *Atg5*FL/FL and the littermate controls *LysM*Cre/+/*Atg5*FL/+, herein called *Atg5*FL/+. By using the FLICA assay, we confirmed the decreased Casp1 activation by *L.g.*+ in *Atg5*FL/+ BMDMs, but this difference was abolished in *Atg5*FL/FL cells (Fig. 6c, d). Similar results were obtained when we measured IL-1β production (Fig. 6e). In accordance with our hypothesis, IFN-β stimulation restored the inhibition of inflammasome activation by *L.g.*− in *Atg5*FL/+ BMDMs, but not in the absence of autophagy (*Atg5*FL/FL) (Fig. 6e). We next assessed a possible LRV-mediated effect in mitophagy[52]. By using BMDMs lacking *Parkin*, an E3-ubiquitin ligase critical for the mitophagic process (*Prkn*−/−)[53], we found that LRV-mediated inhibition of the inflammasome is mitophagy-independent (Fig. 6f). We next used *Atg5* floxed cells to investigate the effect of autophagy in LRV-induced degradation of NLRP3 and ASC, and found that this process requires autophagy. Western blot using anti-ATG5 antibody confirms ATG5 deletion in *Atg5*FL/FL BMDMs (Fig. 6g). Finally, we assessed the intracellular replication of *L.g.*+ and *L.g.*− in *Atg5*FL/+ and *Atg5*FL/FL BMDMs and found that LRV

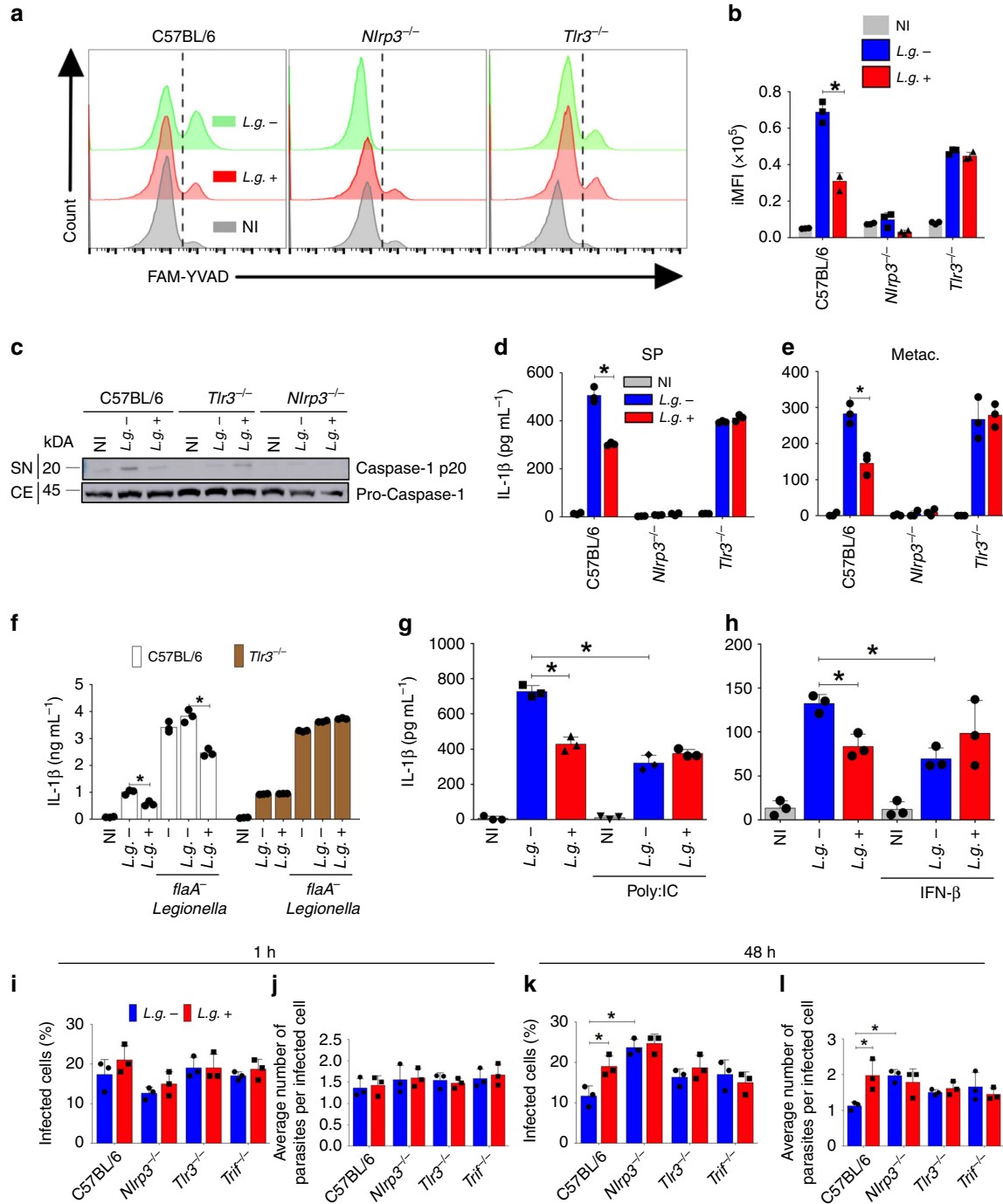

**Fig. 4** LRV triggers TLR3 to limit inflammasome activation by *L.g.* **a**, **b** FLICA analysis showing Casp1 activation (FAM-YVAD) of C57BL/6, *Nlrp3*$^{-/-}$ and *Tlr3*$^{-/-}$ BMDMs infected with either *L.g.*− or *L.g.*+ for 24 h, at a MOI of 10. A representative histogram (**a**) and the integrated MFI (iMFI) (**b**) are shown. **c** Western blotting for cleaved Casp1 (p20) in the SN, and pro-caspase-1 and in the cellular extracts (CE). **d**, **e** ELISA assay for IL-1β in the cell-free supernatants of LPS-primed BMDMs after 24 h of infection with MOI 10 stationary-phase (SP) (**d**) or MOI 5 metacyclic (Metac., **e**) parasites. **f** C57BL/6 (WT) and *Tlr3*$^{-/-}$ BMDMs were infected with *L.g.*− or *L.g.*+ at a MOI of 10, and after 20 h of infection, were infected with *L. pneumophila* (*flaA*− *Legionella*). Four hours later, supernatants were collected and levels of IL-1β were assessed by ELISA. **g**, **h** C57BL/6 BMDMs were infected with *L.g.*− or *L.g.*+ at a MOI of 10, and treated with Poly:IC (5 μg/mL) (**g**) or IFN-β (1000 U/mL) (**h**) at the moment of infection. Twenty-four hours later, supernatants were collected and ELISA for IL-1β was performed. **i**-**l** C57BL/6, *Nlrp3*$^{-/-}$, *Tlr3*$^{-/-}$, and *Trif*$^{-/-}$ BMDMs were infected with metacyclic promastigotes from either *L.g.*− or *L.g.*+, at a MOI of 1. After 1 h of infection, cells were washed and left in culture for 1 or 48 h. Killing of the parasites was evaluated by Panotico *Giemsa* staining. **i** Percentage of infected BMDMs 1 h after infection; **j** average number of amastigotes per cell 1 h after infection; **k** percentage of infected BMDMs 48 h after infection; **l** average number of amastigotes per cell 48 h after infection. The results are shown as mean ± SD. Statistical analysis was performed by unpaired Student's *t* test, and *P* < 0.05 (*) was considered statistically significant. One representative of at least two independent experiments performed in technical triplicates is shown

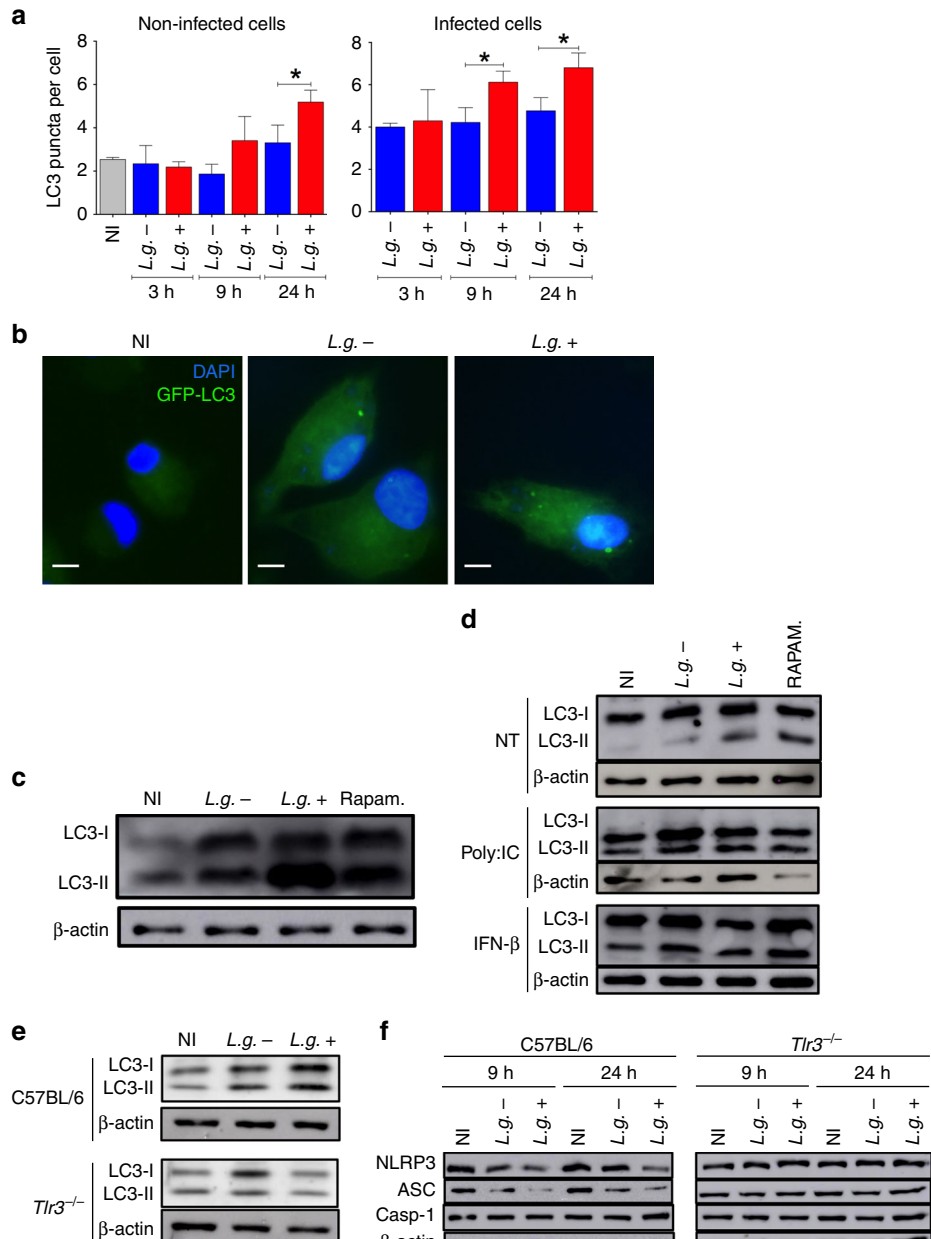

**Fig. 5** LRV increases autophagy via TLR3 and degrade NLRP3 and ASC. **a** Immunofluorescence of C57BL/6 BMDMs transduced with a retroviral vector encoding LC3-GFP, left non-infected (NI) or infected with *L.g.*− or *L.g.*+ for 3, 9, and 24 h. Cells were stained with DAPI (nuclei) and the average number of LC3 puncta per non-infected or infected cell was counted. **b** Representative fluorescence images of non-infected (NI) or infected BMDMs for 24 h. Scale bar: 5 µM. **c** Western Blotting (WB) for LC3-I, LC3-II, and β-actin (loading control) in C57BL/6 BMDMs non-infected (NI) or infected for 24 h with *L.g.*− or *L.g.*+. Rapamycin (1 µM) was used as a positive control. **d** BMDMs were infected and treated with Poly:IC (5 µg/mL) or IFN-β (1000 U/mL) at the moment of infection, and left in culture for 24 h. WB for LC3-I, LC3-II, and β-actin (loading control) was performed. Rapamycin (1 µM) was used as a positive control. **e** C57BL/6 and *Tlr3*−/− BMDMs were infected with *L.g.*− or *L.g.*+ and WB analysis for LC3 was performed after 24 h of infection. **f** After 9 or 24 h of infection, the expression of NLRP3, ASC and Casp1 in NI and infected macrophages was evaluated (F). β-actin was used as a loading control. The results (**a**) are shown as mean ± SD. Statistical analysis was performed by unpaired Student's *t* test, and $P < 0.05$ (*) was considered statistically significant. One representative of at least two independent experiments performed is shown. β-actin was used as a loading control. Data shown in (**a**) is the average of technical triplicates

inhibits intracellular killing of *L. guyanensis*, and this process requires autophagy (Fig. 6h, i). Collectively, these data indicates that autophagy is required for LRV-mediated inhibition of the inflammasome and enhanced parasite's survival within macrophages.

**_L.g._+ derived EVs regulate inflammasome activation.** LRV was shown to be released in extracellular vesicles (EVs) derived from

*L. guyanensis*[39,54], which prompted us to investigate whether vesicle release by the parasite could account for the reduced inflammasome activation observed upon *L.g.*+ infection. Initially, we purified EVs as previously described[55], from both clones (*L.g.*− and *L.g.*+) and confirmed the presence of LRV within vesicles from *L.g.*+, but not *L.g.*−, as expected (Fig. 7a). We found that the profile of EVs secreted by each clone is similar, with the mean/mode diameter of most vesicles ranging from 50 to

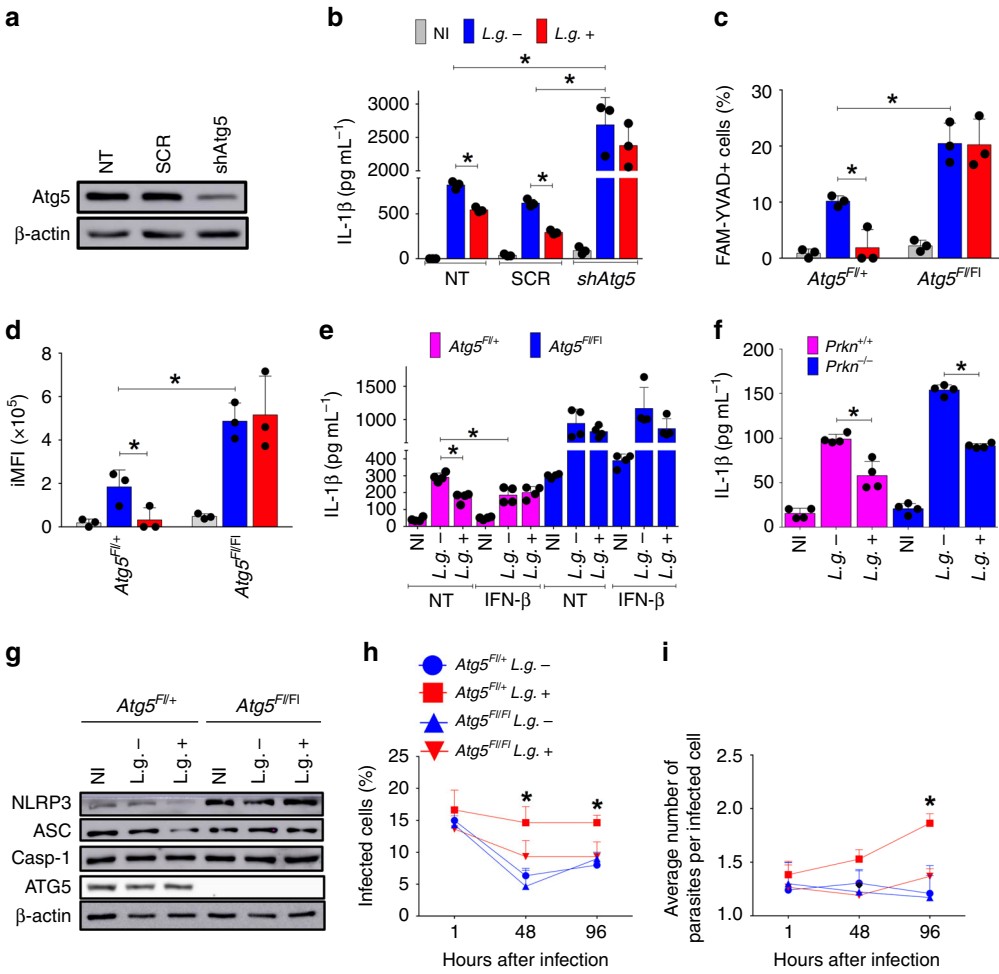

**Fig. 6** LRV limits NLRP3 activation via ATG5. **a**, **b** C57BL/6 BMDMs were silenced with a scramble sequence (SCR) or *Atg5*-specific shRNA, then infected with *L.g.*− or *L.g.*+, and the efficiency of the process (**a**) was confirmed by WB. β-actin: loading control. **b** After 24 h of infection, levels of IL-1β were determined by ELISA in cell-free supernatants. NT: Non-transduced BMDMs. **c**, **d** BMDMs from *LysM^{Cre/+}/Atg5^{Fl/+}* (Littermate control) or *LysM^{Cre/+}/Atg5^{Fl/Fl}* (Knockout) mice were infected with *L.g.*− or *L.g.*+ for 24 h, and Casp1 activation was determined by FLICA assay (FACS). The percentage of Casp1 + cells (FAM-YVAD) (**c**) and iMFI (**d**) are shown. **e** *LysM^{Cre/+}/Atg5^{Fl/+}* and *LysM^{Cre/+}/Atg5^{F/Fl}* BMDMs were infected and left untreated or treated with IFN-β (1000 U/mL) at the moment of the infection. Levels of IL-1β in cell-free SNs were quantified by ELISA, after 24 h of infection. **f** ELISA assay in Parkin deficient BMDMs (*Prkn^{−/−}*) and its respective littermate control (*Prkn^{+/+}*) after 24 h of infection. **g** WB for inflammasome components and ATG5 in *LysM^{Cre/+}/Atg5^{Fl/+}* and *LysM^{Cre/+}/Atg5^{Fl/Fl}* BMDMs after 24 h of infection with *L.g.*− or *L.g.*+. β-actin was used as a loading control. **h**, **i** BMDMs were infected with metacyclic promastigotes from either *L.g.*− or *L.g.*+, at a MOI of 1. One hour after infection, cells were washed and left in culture for 1, 48, or 96 h. The percentage of infected cells (**h**) and the average number of amastigotes per cell (**i**) was evaluated by Panotico *Giemsa*. The results are shown as mean ± SD. Statistical analysis was performed by unpaired Student's *t* test. *P* < 0.05 (*) was considered statistically significant. One representative of at least two independent experiments performed with technical replicates is shown

200 nm, suggestive of exosomes[56] (Fig. 7b). We then treated different subsets of BMDMs with EVs derived from both clones, at the moment of the infection. Strikingly, while *L.g.*− -derived EVs did not affect IL-1β production upon infection by both clones, *L.g.*+-derived EVs rescued *L.g.*− capacity to impair inflammasome activation in WT macrophages in a TLR3-dependent manner (Fig. 7c). Interestingly, the same effects were observed for *LysM^{Cre/+}/Atg5^{FL/+}* cells, but not for *LysM^{Cre/+}/Atg5^{FL/FL}* (Fig. 7d). Next, we tested if incubation of *L.g.*− parasites with EV derived from *L.g.*+ cells can complement LRV + functions in inhibiting inflammasome activation. Thus, we treated stationary-phase promastigotes with *L.g.*+-derived EVs as previously described[39]. We observed that after washings, LRV genetic material was detected in cultures incubated with *L.g.*+-derived EVs (Fig. 7e). Importantly, *L.g.*− parasites that were incubated with *L.g.*+-derived EVs (*L.g.*- EVs LRV +) induced less inflammasome activation than the parental *L.g.*− strain,

suggesting that LRV transferred via EV complemented the effects in inflammasome inhibition (Fig. 7f). Next, we tested if LRV + EVs rescue *L.g.*− defects in intracellular replication in macrophages. By scoring intracellular parasites after 48 h infection of C57BL/6 BMDMs, we found that *L.g.*− EVs LRV + replicated better than *L.g.*−, suggesting that *L.g.*+ -derived EVs are able to complement the LRV effects in *L.g.*− strain (Fig. 7g-j). Taken together, these results suggest that EVs obtained from *L.g.*+ strains are able to complement the *L.g.*− defects in inhibition of inflammasome activation and intracellular replication in macrophages.

**NLRP3 and ATG5 are required for LRV effects in vivo.** Our data reveal a mechanism induced by LRV during *L.g.* infection, in which the virus triggers TLR3 signaling to induce autophagy, that inhibits inflammasome activation and enhances parasite's survival in BMDMs. To investigate whether these in vitro findings are

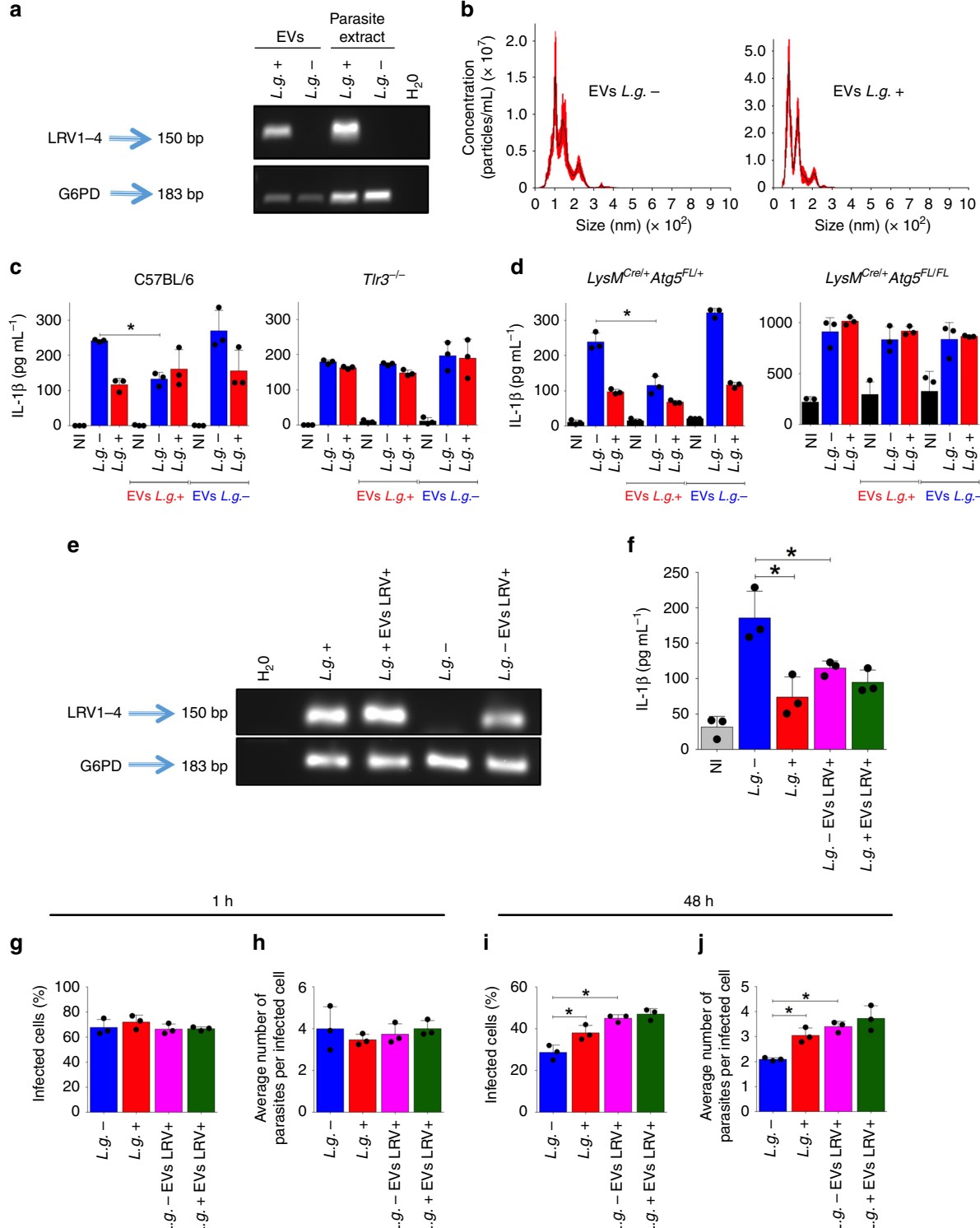

**Fig. 7** LRV is shed within EVs and complements *L.g.*−. **a** LRV and G6PD were detected by PCR in extracellular vesicles (EVs) purified from *L.g.*− and *L.g.*+ (EVs *L.g.*− and EVs *L.g.*+). Parasite extracts (*L.g.*− and *L.g.*+) and H₂O were used as PCR controls. **b** Nanoparticle-tracking analysis of EVs produced by *L.g.*− and *L.g.*+. **c, d** C57BL/6 and *Tlr3*−/− (**c**), *LysM*Cre/+/Atg5FL/+ and *LysM*Cre/+/Atg5FL/FL (**d**) BMDMs were primed with LPS and infected with stationary-phase parasites (MOI 10). At the moment of infection, cells were treated with EVs purified from *L.g.*− or *L.g.*+. After 24 h, IL-1β levels were determined by ELISA in cell-free supernatants. NI: Non-infected. **e, f** Stationary-phase promastigotes of *L.g.*−, or *L.g.*+ (used as control), were incubated for 2 h with EVs derived from *L.g.*+ in Schneider's medium (*L.g.*+ EVs LRV + and *L.g.*− EVs LRV +). Parasites were isolated for RNA extraction and PCR analysis (**e**) or used for BMDMs infection at MOI 10, to measure IL-1β production (**f**). **g–j** C57BL/6 macrophages were infected with *L.g.*− or *L.g.*+ (incubated or not with EVs from LRV +) at MOI 5 for replication assay. After 1 (**g, h**) or 48 h (**i, j**), the percentage of infected cells and the average number of parasites per cell were estimated. The results are shown as mean ± SD and statistical analysis was performed by unpaired Student's *t* test. *P* < 0.05 (*) was considered statistically significant. One representative of at least two independent experiments performed with technical replicates is shown

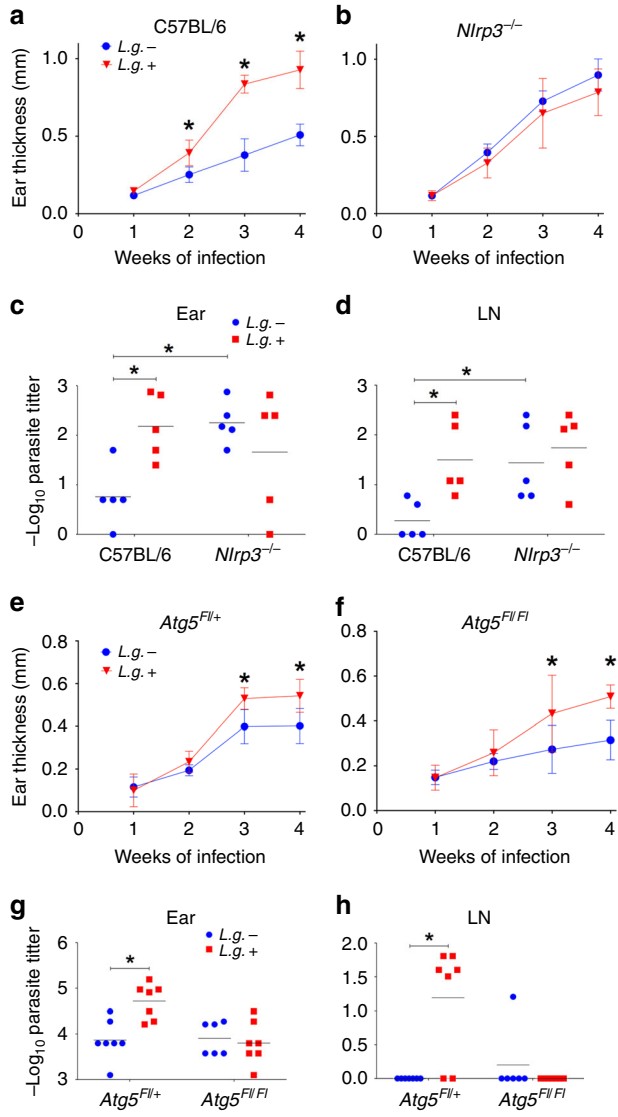

**Fig. 8** LRV exacerbates *L.g.* infection via NLRP3 and ATG5 in vivo. **a** C57BL/6 and **b** *Nlrp3*$^{-/-}$ mice were injected with $10^5$ Metacyclic promastigotes of *L.g.*- or *L.g.*+ (*n* = 5 mice per group), and ear thicknesses were followed weekly. Four weeks after infection, parasite titers were determined in the ear (**c**) and draining lymph node (**d**). One representative of three independent experiments performed is shown. LysM$^{Cre/Fl}$Atg5$^+$ (**e**) and LysM$^{Cre/Fl}$Atg5$^{Fl}$ (**f**) mice were injected with $10^5$ Metacyclic promastigotes of *L.g.*− or *L.g.*+ and ear thicknesses were followed weekly. Four weeks after infection, parasite titers were determined in the ear (**g**) and draining lymph node (**h**). The results are shown as mean (**c**, **d**, **g**, **h**) or mean ± SD (**a**, **b**, **e**, **f**) from data pooled from two independent experiments (*n* = 6-7 mice per group). Statistical analysis was performed by two-way ANOVA with Bonferroni's multiple comparison test. *P* < 0.05 (*) was considered statistically significant

physiologically relevant during in vivo infection, we infected C57BL/6 and *Nlrp3*$^{-/-}$ mice with *L.g.*− or *L.g.*+. Confirming our previous experiments, *L.g.*+ infection induced increased ear thickness compared with *L.g.*− in C57BL/6 (Fig. 8a and Supplementary Fig. 8a), but not in *Nlrp3*$^{-/-}$ (Fig. 8b), *Asc*$^{-/-}$ (Supplementary Fig. 8b), *Casp1*$^{-/-}$ (Supplementary Fig. 8c) or *Tlr3*$^{-/-}$ (Supplementary Fig. 8d) mice. Accordingly, increased parasite titers of *L.g.*+ were observed in the ear (Fig. 8c) and lymph node (Fig. 8d) of wild-type infected animals, but not in NLRP3 (Fig. 8c, d), ASC, CASP1 or TLR3-deficient mice

(Supplementary Fig. 8e, f). Moreover, we infected *Atg5*$^{FL/+}$ (*LysM*$^{Cre/+}$/*Atg5*$^{FL/+}$) and *Atg5*$^{FL/FL}$ (*LysM*$^{Cre/+}$/*Atg5*$^{FL/FL}$) animals with both *L.g.* clones. Although ear thicknesses did not differ between infected-littermate controls (Fig. 8e) and knockout animals (Fig. 8f), *Atg5*$^{FL/+}$ mice displayed increased parasite titers in the ear (Fig. 8g) and lymph node (Fig. 8h) upon *L.g.*+ infection, while *Atg5*$^{FL/FL}$ did not.

**LRV limits NLRP3 activation by *Lg* in human macrophages.**
Next, we tested whether LRV signaling also dampens inflammasome activation in human monocyte-derived macrophages. We differentiated highly pure CD14$^+$-monocytes obtained from different healthy donors into macrophages by culturing these cells in the presence of GM-CSF for 7 days, and measured inflammasome activation upon *L.g.* infection. We found that both *L.g.*+ and *L.g.*− induce IL-1β (Fig. 9a) and Casp1 cleavage (Fig. 9b) in human macrophages. As observed in mouse cells, *L.g.*+ induced less inflammasome activation in human macrophages than *L.g.*− (Fig. 9a, b). We primed human macrophages with the TLR2 agonist PAM(3)CSK(4) since LPS is known to induce TLR4-dependent activation of the inflammasome in human monocytes[57]. Next, we tested whether LRV affects autophagy induction in human cells. Similar to our data using BMDMs, infection of human macrophages with *L.g.*+ induced autophagy more robustly than *L.g.*− (Fig. 9c, d). To further investigate the LRV-inhibition of the NLRP3 inflammasome in human cells, we measured IL-1β and Casp1 p20 in response to *L.g.*− and *L.g.*+ infection in cells treated with KCl (which impairs potassium efflux and activation of NLRP3) or Poly:IC (TLR3 agonist). NaCl was used as a control for KCl treatment. Our data indicates that LRV limits NLRP3 activation and Poly:IC rescue the inhibitory effect of LRV in cultures infected with *L.g.*− (Fig. 9e, f). As expected, KCl, but not NaCl treatment inhibited IL-1β and Casp1 p20 production in response to infection, confirming the participation of the NLRP3 inflammasome in human cells (Fig. 9e, f). Although the treatment with KCl and Poly:IC did not interfere with parasite internalization (Fig. 9g, h), both treatments rescued *L.g.*− phenotype for intracellular replication, as observed for *L.g.*+. This is shown by both the percentage of infected cells (Fig. 9i) and the average number of parasites per infected cell (Fig. 9j). Taken together, these results expand to human cells our mechanistic findings regarding the biology of LRV during *Leishmania* infection.

**LRV impairs NLRP3 activation by isolates of *Lb* and *Lg*.** Our mechanistic studies performed so far employed the strain M4147 (*L.g.*+) and the clone 40 (*L.g.*−) that spontaneously lost LRV. To expand the number of parasites tested, we obtained three additional strains of *L. guyanensis* from FIOCRUZ *Leishmania* collection (Supplementary Table 2, IOCL3539 isolate is LRV+ whereas IOCL3460 and IOCL3538 are LRV−) and tested for inflammasome activation. We found that the clinical isolate that is LRV+ induced similar levels of IL-1β production to *L. guyanensis* M4147 (*L.g.*+), and significantly less than the clone 40 (*L.g.*−) (Fig. 10a). By contrast, the LRV− strains (IOCL3460 and IOCL3538) were similar to clone 40 and induced significantly more IL-1β production than *L.g.*+ and IOCL3539 (Fig. 10a).

In addition to *L. guyanensis*, other New-World species of the *Viannia* subgenus, including *L. braziliensis*, can also harbor LRV[37]. Thus, we obtained 20 clinical isolates of *L. braziliensis* from FIOCRUZ *Leishmania* (Supplementary Table 2) and assessed IL-1β secretion upon macrophage infection. We cultivated and synchronized all 20 isolates and performed the infections simultaneously. We found that inflammasome activation is significantly reduced in response to infection with the 12

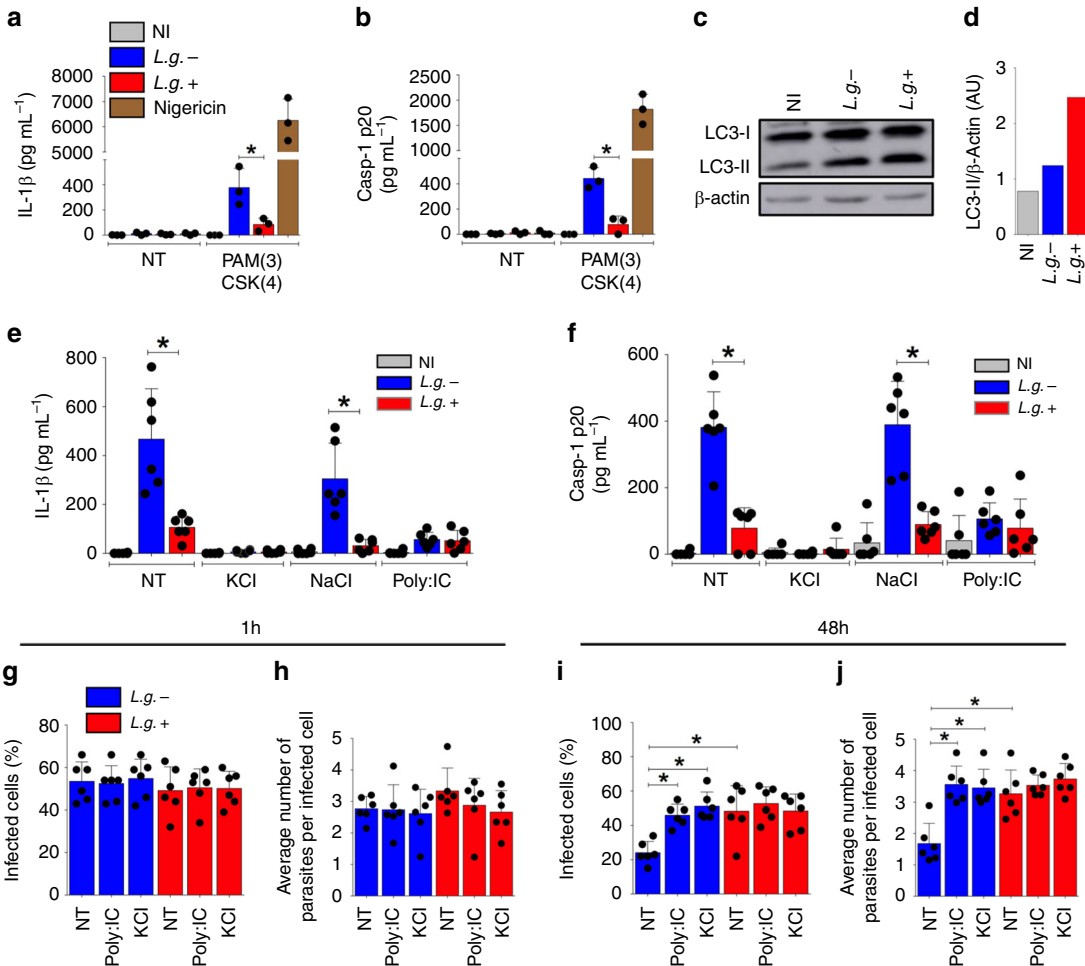

**Fig. 9** LRV modulates NLRP3 inflammasome in human macrophages. Human monocyte-derived macrophages were left unprimed or primed with PAM(3) CSK(4) (300 ng mL$^{-1}$) for 4 h, and then infected with *L.g.*− or *L.g.*+. The levels of IL-1β (**a**) and Casp1 p20 (**b**) were determined in cell-free supernatants by ELISA. **c** Human macrophages from three different donors were mixed, plated and left non-infected (NI) or infected with *L.g.*− and *L.g.*+. The expression of LC3-I and LC3-II was measured by western blotting. **d** Quantification of LC3-II was performed by dividing LC3-II densitometry values provided by ImageJ per β-actin (arbitrary units, AU). **e**, **f** PAM(3)CSK(4)-primed human macrophages were left untreated (NT) or treated with Poly:IC (5 µg/mL), NaCl (100 mM), or KCl (100 mM) at the moment of infection. The levels of IL-1β (**e**) and Casp1 p20 (**f**) were determined in cell-free supernatants by ELISA. **g**–**j** Unprimed human macrophages were treated as described above to evaluate parasite killing by Panotico *Giemsa*. The percentage of infected cells (**g**, **i**) and the average number of parasites per infected cell (**h**, **j**) is shown at 1 and 48 h after infection. The results are shown as mean ± SD. Statistical analysis was performed by unpaired Student's *t* test, while $P < 0.05$ (*) was considered statistically significant. One representative of at least two independent experiments performed is shown. Each dot in graphs represents macrophages from an individual donor. The results are demonstrated as the average of biological replicates

clinical isolates that are LRV+ as compared with the eight LRV− isolates (Fig. 10b). Taken together, these data indicate that regardless of the species and individual variations, the presence of LRV in New-World *Leishmania* influences inflammasome activation by a TLR3/TRIF/IFN-β/ATG5 axis, which is critical for the outcome of Leishmaniasis (Fig. 10c).

## Discussion

Despite the remarkable importance of Leishmaniasis, a disease that affects millions of people worldwide, the cellular and molecular mechanisms responsible for pathogenesis are not fully characterized and many aspects of the disease remain unknown[4]. In this study, we demonstrate an evasion mechanism triggered by New-World *Leishmania* species harboring the endosymbiont dsRNA virus, which is an important risk factor for the development of mucocutaneous disease[37]. Although LRV contributes to disease severity, other risk factors, such as co-infections, host immune responses and individual variations of the parasite

strains may also be critical to determine clinical outcome of Leishmaniasis in patients. Regardless of additional factors, in the current study we identified that LRV is found within EVs and promotes TLR3-mediated induction of autophagy through type I IFN. This process results in reduced inflammasome activation through autophagy-mediated degradation of specific inflammasome components. We hypothesize that LRV-mediated inhibition of inflammasome activation at early stages of infection results in ineffective parasite elimination in the tissues. As a consequence, robust T cell activation could promote the exacerbated and unregulated immune response observed in the severe mucocutaneous forms of the disease[58–64]. Interestingly, while inhibiting inflammasome activation through autophagy, our data indicates that LRV-mediated TLR3 activation also favors the induction of inflammatory cytokines, such as TNF-α and IL-12 (Supplementary Fig. 3 and Fig. 10c). This is in agreement with previous data indicating that TLR3 recognizes LRV and worsens the disease through induction of inflammatory cytokines[36].

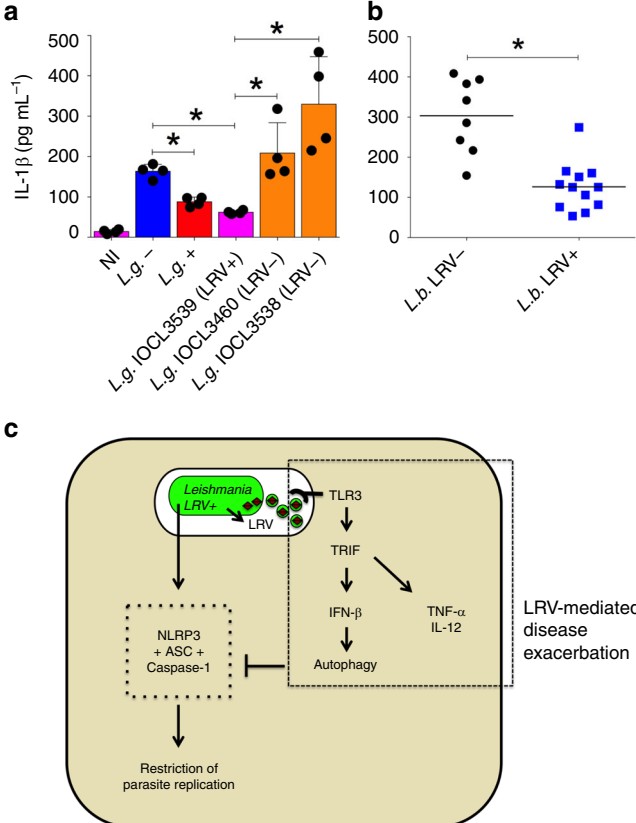

**Fig. 10** LRV+clinical isolates induces less inflammasome activation. **a** LPS-primed bone marrow-derived macrophages (BMDMs) were left uninfected (NI) or infected (MOI 10) with *L.g.*− (blue bar), *L.g.*+ (red bar) or different *L. guyanensis* clinical isolates. Two of them are LRV− (orange bars) and one is LRV + (pink bar). **b** LPS-primed BMDMs were infected with 20 different clinical isolates of *L. braziliensis* (MOI 10) obtained from different patients. Isolates that are LRV− are shown as circles (black circles) and isolates LRV + are shown in as squares (blue squares). Information of each isolate is contained in Supplementary Table 2. After 24 h of infection, cell-free supernatants were collected and the levels of IL-1β were measured by ELISA. The average value for each clinical isolate was plotted together, according to LRV presence/absence. The results are shown as mean ± SD. Statistical analysis was performed by unpaired Student's *t* test, with *P* < 0.05 (*) considered statistically significant. One representative of three independent experiments performed with technical replicates is shown. **c** The NLRP3 inflammasome is important to control *Leishmania* in vitro and in vivo. Parasites harboring an endosymbiontic dsRNA virus (LRV) display increased virulence and represent a risk factor for the development of the mucocutaneous form of Leishmaniasis. Mechanistically, LRV (in brown) transits within extracellular vesicles (green circles) in the vacuole, eventually triggering TLR3/TRIF signaling to induce several inflammatory cytokines, such as pro-IL-1β, TNF-α, and IL-12. While increasing inflammation, LRV also induces IFN-β, which activates the autophagic machinery to degrade NLRP3 and ASC, limiting NLRP3 activation and favouring *Leishmania* replication

that we successfully obtained an isogenic clone of M4147 without the virus (*L.g.*−) regardless of genetic manipulation of the parental strain. This provided us with a powerful tool to address the effects of the virus in the pathogenesis and induction of innate immune responses. Of note, a previous study from Hartley and colleagues did not detect any activation of the inflammasome when comparing *Leishmania guyanensis* with and without the virus[41]. Although inflammasome activation in *Leishmania*-infected macrophages has been previously reported by many independent groups[25–27,29,42,43,65,66], it is a consensus in the field that the magnitude of inflammasome activation in response to *Leishmania* is low when compared with other microbes, including other protozoan parasites[67]. In agreement to these observations, it has been suggested that different species of *Leishmania* actively inhibits inflammasome activation[66,68,69], a feature that is consistent with the important role of the inflammasome for restriction of parasite replication[27,42,43,68,70,71]. In this scenario, we envisage that specific macrophage infection conditions used by Hartley and colleagues did not allow detection of any inflammasome activation[41]. Consequently, they did not identify the effect of LRV+ parasites in inflammasome inhibition when compared with the LRV− strain of *L. guyanensis*.

Although NLRP3 activation by *Leishmania* has been investigated in murine models of infection[25–27,42,71], we are unaware of any prior reports comparing the magnitude of inflammasome activation in patients with different forms of the disease. While some groups reported a deleterious effect of NLRP3 and IL-1R signaling to the host[25,26,65], others have shown a protective role of the inflammasome during infection[27,42,43,71]. Now we demonstrate an inverse correlation between the severity of Leishmaniasis and inflammasome activation, supporting a protective role of the inflammasome during Leishmaniasis. However, it is possible that the inflammasome may favor immunopathology and/or parasite control depending on the course of infection, parasite strain and genetic factors associated with the host. Importantly, our data with human macrophages and clinical samples from LRV+ and LRV− patients, together with *L.g.*+ and different *Leishmania* isolated from humans, demonstrate that LRV dampens NLRP3 activation to favor infection and pathogenesis.

Several groups have reported the mechanisms driving NLRP3 activation by *Leishmania*, but the mechanisms regulating its inhibition are still poorly known. Although TLRs are thought to provide the first signal required to activate inflammasomes, our results obtained with TLR3-deficient cells and with Poly:IC show that TLR3 is triggered by LRV to inhibit NLRP3 activation by *L.g.* Differently from all other TLRs, TLR3 signals exclusively via TRIF, triggering potent type I IFN production[15,72], a process that was previously reported to inhibit inflammasome activation[49]. Likewise, type I IFN was previously shown to induce autophagy[50]. Type I IFN has long been attributed a deleterious role to the host during Leishmaniasis, albeit the mechanisms responsible for that are unclear[19,20,40]. Here, we link TLR3 and IFN-β with inhibition of inflammasome activation through autophagy, providing a possible explanation for the previously reported detrimental effects of type I IFN during *Leishmania* infection. These data contributes to advance our understanding of the effects from type I IFN during Leishmaniasis.

*Leishmania* infection has been reported to trigger autophagy, but its impacts in the outcome of the infection may vary, depending on the parasite species, mouse background, and other factors[18,31–33,73,74]. Here, we report that autophagy is triggered by *L.g.*− and regulates inflammasome activity, but plays no significant role in parasite replication. However, TLR3 signaling by LRV boosts autophagy to degrade NLRP3 and ASC, promoting parasite survival. Alongside, LRV boosts inflammatory cytokines, such as TNF-α and IL-12. It is possible that LRV-acquisition

A few studies reported the host detrimental effects of LRV during infection and specific mechanisms involving induction of TLR3 activation and type I IFN production[34,36,38]. One particular study evaluated the impact of LRV in the course of *Leishmania* infection comparing two independent clinical isolates, one that promotes metastasis (*L.g.*M+) and the other non-metastasizing (*L.g.*M−)[36]. Other studies compared M4147 (LRV^high) and a mutated HygromicinB-resistant virus-free derived clone[36,38,39,52–54]. One important aspect of our study is the fact

during evolution promoted a selective advantage in mammalian hosts by inhibiting inflammasome-mediated restriction of parasite replication. Nonetheless, the virus also promotes TLR3-mediated production of inflammasome-independent inflammatory cytokines, which may ultimately contribute to the development of mucocutaneous lesions in certain patients.

Taken together, our findings combine data obtained from clinical samples and mice to mechanistically demonstrate an evasion mechanism triggered by LRV to promote *Leishmania* pathogenesis. This work also sheds light into previously unappreciated cross-talks between different innate immune sensors and processes, pointing to autophagy as an important regulator of inflammasome activation during *Leishmania* infection. This work contributes to our global understanding of specific risk factors related to the development of mucocutaneous Leishmaniasis in humans. In addition, it provides specific targets for the development of different therapeutic strategies. Future development of drugs that target specific pathways triggered by this virus might be useful for better clinical outcomes in patients suffering from highly debilitating mucocutaneous Leishmaniasis.

## Methods

**Animals**. Mice used in this study were in C57BL/6 genetic background and included C57BL/6 (JAX, stock number 000664), $Trif^{-/-}$ (JAX, stock number 005037), $Nlrp3^{-/-}$[75], $Asc^{-/-}$, $Casp1/11^{-/-}/Casp11^{Tg}$ (herein called $Casp1^{-/-}$)[76], $Prkn^{-/-}$[53] and $Tlr3^{-/-}$[16]. $Atg5^{Fl/Fl}$ and $Atg5^{Fl/+}$ mice[77] were breed with $LysM^{Cre/+}$ mice (JAX) to generate $LysM^{Cre/+}/Atg5^{Fl/Fl}$ (herein called $Atg5^{Fl/Fl}$) and the littermate controls $LysM^{Cre/+}/Atg5^{Fl/+}$ (herein called $Atg5^{Fl/+}$). Female mice ranging from 6- to 8-weeks-old were bred and maintained under specific pathogen-free conditions at the University of São Paulo, FMRP/USP animal facility. All animals were provided food and water ad libitum, at 25 °C. The care of the mice was in compliance with the institutional guidelines on ethics in animal experiments; approved by CETEA (Comissão de Ética em Experimentação Animal da Faculdade de Medicina de Ribeirão Preto, approved protocol number 014/2016). CETEA follow the Brazilian national guidelines recommended by CONCEA (Conselho Nacional de Controle em Experimentação Animal). We declare, in the most emphatic terms, that we have complied with all relevant ethical regulations for animal testing and research.

**Parasite culture and infection in vivo and in vitro**. The wild-type strain of *L. guyanensis* M4147 (MHOM/BR/75/M4147), which harbors high levels of LRV1, and derivative clones positive or negative for the virus (*L.g.−*, described below), were used for both in vitro and in vivo experiments. Parasites were cultured at 25 °C in Schneider's Drosophila medium (Invitrogen, Carlsbad, CA), pH 7.0, supplemented with 10% heat-inactivated fetal calf serum (GIBCO BRL), 2 mM L-glutamine, and 2% urine, pH 6.5. For in vivo infections, mice were infected with either $1 \times 10^6$ stationary phase or $1 \times 10^5$ metacyclic promastigotes in 10 μL of PBS, through an intradermal injection into the left ear. The infective-stage metacyclic promastigotes of *L. guyanensis* were isolated from stationary cultures through density gradient centrifugation[78]. Ear thicknesses were monitored weekly with a dial gauge caliper and compared with the thickness of the uninfected contralateral ear. Parasite burdens were determined in the ear and retromaxilar lymph node, which drains the site of infection[79]. The parasites used in each experiment were obtained from liquid nitrogen (frozen stock) or directly from infected mice. In all experiments, the parasite cycle were synchronized by passaging (cultivating in axenic media) twice in log phase (2 days of growth in fresh Schneider's medium, $10^5$ parasites/mL). After that, parasites are cultivated in axenic media for the infection (also $10^5$ parasites/mL of Schneider) and we wait until they reach stationary-phase (normally 5 days cultivation in axenic phase). Each passage of the parasite in axenic media were monitored and used for experiments. After the 6th passage the parasites were discarded. After infection, plates were centrifuged 1200 rpm for 5 min to allow synchronized entry of the parasites into BMDMs. For the experiments that do not use the cell-free supernatants (*Giemsa* and FACS) we infect cultures for 1 h and then wash each well twice with fresh 1X PBS and replace with fresh RPMI plus 10% FBS. In assays that use the supernatants (ELISA, western blotting for Casp1 p20 and others) the infected cultures were not washed after infection[42,43].

**Human monocytes purification and differentiation**. Total blood was collected from healthy donors under their consent (Ethical committee protocol number 18492/2014) in EDTA-coated tubes (BD Vacutainer CPT™) according to the manufacturer's instructions, and centrifuged at $400 \times g$ for 10 min at room temperature. Then, plasma was discarded and the pellet was diluted in PBS 1X pH 7.4 (GIBCO, BRL). Cells were applied in Ficoll-Paque™ PLUS gradient (GE Healthcare Biosciences AB, Uppsala, Sweden). Afterward, they were centrifuged at

$600 \times g$ for 30 min at room temperature in order to obtain the purified mononuclear fraction of cells, which were carefully collected and transferred to a different tube. After that, cells were washed twice and the pellet was ressuspended in PBS.

Mononuclear cells were incubated with CD14-coupled magnetic beads (MiltenyiBiotec, Auburn, CA, EUA) and then applied in a column for magnetic separation, according to the manufacturer's instructions. The resultant population of purified monocytes (CD14+) was confirmed by FACS (ACCURI C6, BD Biosciences) and cultivated in RPMI 1640 (GIBCO, BRL) supplemented with 10% FBS and 50 ng/mL of human GM-CSF (R&D Systems). After 7 days, monocytes were differentiated and ready to use for in vitro experiments.

**Recruitment of patients**. Samples were collected from 49 invited-patients at the Rondônia Reference Hospital for Tropical Medicine, CEMETRON. They were diagnosed with Leishmaniasis from June to November 2017. CEMETRON Hospital is located in Porto Velho and is a reference center for patient care.

**Human ethics statement**. The ethical recommendations of the Brazilian National Council of Health were followed with a research protocol registered and approved under the Certificate of Presentation for Ethics Appreciation code (CAAE, Number 54386716.1.0000.0011) by the Ethical and Research Committee of the Center of Research in Tropical Medicine (CEP/CEPEM). Patients who voluntarily agreed to participate in the research were appropriately informed and signed a consent form. Patients were diagnosed according to the recommendations of the Brazilian Ministry of Health, routinely applied at the Hospital, which includes clinical and epidemiological criteria. The clinical manifestation of Leishmaniasis was classified as Cutaneous Leishmaniasis (CL) or Mucocutaneous Leishmaniasis (MCL). Samples from 49 patients diagnosed with Leishmaniasis (40 CL and 9 MCL) were included in the study. All patients were tested for HIV and samples used in this study were from HIV negative patients.

**Human Samples collection and processing**. Sample collection was performed using a sterile cervical brush placed in direct contact with the internal edge of the lesions. In CL patients, cervical brushes were obtained from cutaneous lesions, while in MCL patients samples were acquired from nasal tissues. Sampling was performed for both RNA extraction and cytokine/protein quantification. The collected material was immediately stored in an RNAlater solution (Ambion, Austin, TX, USA) for preservation of molecular contents and was stored at −20 °C until the time of analysis.

The RT-PCR technique was applied for LRV1 detection. Total RNA was extracted using the PureLink RNA Mini Kit (Invitrogen, Carlsbad, CA, USA), and reverse transcription was obtained using SuperScriptIII (Invitrogen, Carlsbad, CA, USA) following the manufacturer's recommendations. The cDNA was submitted to PCR to amplify a 240 bp fragment (LRV1F–5′-ATGCCTAAGAGTTTGGATTCG-3′/LRV1R–5′ ACAACCAGACGATTGCTGTG-3′) corresponding to the ORF1 region of the viral genome. The forward primer matches exactly the previously described region. The reverse primer was designed using Primer3 and checked for specificity using Primer-BLAST. The amplification reactions were performed in a final volume of 50 μL containing 3 μL of cDNA, 0.2 mM of each primer, 200 μM dNTP, 0.75 mM $MgCl_2$ and 1 U Taq DNA Polymerase Invitrogen (Carlsbad, CA, USA) using the following cycling: 94 °C for 5 min, followed by 30 cycles at 94 °C for 20 s, 57 °C for 15 s and 72 °C for 30 s, and a final extension at 72 °C for 5 min. The fragments were visualized on a 2% agarose gel. The cDNA obtained from the strain MHOM/BR/1975/M4147, donated by Coleção de Leishmania do Instituto Oswaldo Cruz (CLIOC), was used as a positive control for all reactions. Some amplicons were sequenced using capillary Sanger sequencing standards protocols to check proper and specific amplification.

Protein quantification was performed using the Bradford reagent (Sigma-Aldrich). Levels of different cytokines were measured using BD (IL-1β and TNF-α) or R&D (Casp1 p20) human ELISA kits, and the values obtained for each cytokine was normalized to the concentration of protein (measured by Bradford) from each single patient, being represented as picograms of cytokine per milligram of protein).

**Flow cytometric analysis of *L.g.−* infected BMDMs**. A total of $10^7$ *L.g.−* or *L.g.+* metacyclic parasites were incubated with 1 mL of RPMI medium supplemented with 10% FBS and 10 μM of CFSE (CellTrace™ CFSE Cell Proliferation Kit, Invitrogen). The suspensions were incubated for 1 h at 37 °C and homogenized in every 15 min. Then, parasites were washed with 10 mL of fresh PBS1X (Gibco) and ressuspended with 5 mL of PBS1X, and left for 30 additional minutes at 37° to allow de-esterification of the dye. Finally, parasite suspensions were centrifuged (4000 rpm/10 min) and ressuspended in appropriate volumes for the infection. BMDMs were washed after 1 h of infection and left in culture up to 96 h. The data were acquired on a FACS ACCURI C6 flow cytometer (BD Biosciences) and analyzed with the FlowJo software (Tree Star).

**Legionella pneumophila co-infection**. In the co-infection experiments, BMDMs were infected for 20 h with either *L.g.−* or *L.g.+*, and then co-infected with JR32 *Legionella pneumophila* mutants for flagelin (*flaA- Legionella*) at a MOI of 10, for

additional 4 h, totalizing 24 h of infection. Supernatants were collected and IL-1β was measured by ELISA, as described below.

**Generation of *L.g.*− and LRV+ clones from *L.g.* M4147.** An old culture passage (±50 passages) was used to evaluate if all cells of *L. guyanensis* M4147 carry the LRV1 virus. We performed the isolation of single cells in semisolid medium to obtain isolated colonies according to Iovannisci & Ullman (1983) with some modifications. Promastigotes growing exponentially were washed twice in PBS 1X and diluted to a concentration of 3000 cells/ml into freshly prepared M199 medium and 0.1 ml spread onto semisolid plates, M199 medium supplemented with 1% noble agar (Sigma-Aldrich) to a final expected density of 300 colonies/plate. The plates were incubated with a 5% $CO_2$ atmosphere at 26 °C. *L. guyanensis* colonies were visible after 10 days and every colony were picked to M199 medium containing 10% fetal bovine serum and supplied with 2% fresh filtered urine until reaching late log/early stationary phase. Among all isolated colonies, the clone 40 was the only one negative for LRV1–4 (*L.g.*−). Five random LRV+ clones were selected and used for in vitro experiments.

**LRV1 detection by PCR.** A total of $5 \times 10^7$ promastigotes were washed with phosphate-buffered saline, cells were centrifuged at $2000 \times g$ for 5 min, and resuspended and lysed in 750 μL of Trizol (Life Technologies). Total RNA was isolated according to the manufacturer's instructions. The complementary DNA (cDNA) was prepared from 1 μg of total RNA with random hexamers using the kit *High-Capacity cDNA Reverse Transcription Kit* (Applied Biosystems) according to the manufacturer's instructions. The PCR was performed by 30 cycles, using LRV1 specific primers (LRV1–4 F: 5′-TAATCGAGTGGGAGTCCCCC-3′ and LRV1–4 R: 5′-GATCCTCCACACCGACCGTA-3′) that amplified a 151-nucleotide segment within the LRV1 5′UTR region. *Leishmania* specific primers (G6PD F: 5′-CAGATGGAAGCGTGTGATCG-3′-G6PD R: 5′-TGCGAGCATAGCCGACA-3′) were also used as a control. RT-PCR products were analyzed on a 1.5% agarose gel in 1X TAE buffer.

**LRV1 detection by immunofluorescence.** A total of $8 \times 10^6$ stationary-phase promastigotes cells were washed twice with PBS and fixed with 2% formaldehyde in PBS for 20 min at room temperature. Cells were washed three times with PBS and then the pellets were suspended and incubated with 0.2 M Glycine for more 20 min and washed twice with PBS. The final pellet was resuspended in 300 μL of PBS and spotted in microscope slides for 30 min at RT. The slides were washed twice with PBS for 5 min and cells were permeabilized/blocked with PBS solution with 0.2% Triton 100 × (Sigma-Aldrich), 1% BSA (Fraction V, Sigma), and 5% of goat serum, for 30 min at room temperature. Upon permeabilization, cells were incubated at RT for 1 h with mouse anti-dsRNA J2 monoclonal antibody (1:800, English & Scientific Consulting) in 1% BSA in PBS-TX. Cells were then washed five times in PBS, incubated for 1 h with a FITC-labeled secondary antibody from goat anti-mouse IgG Apl24F (Millipore, Temecula, CA, USA) diluted 1:100, for 1 h in a dark chamber. Cells were washed three times with PBS and incubated 10 min with 0.5 mg/ml 49,6-Diamidino-2-phenylindole (DAPI, Invitrogen), washed again and finally mounted in mounting media (Dako), and stored at 4 °C until analysis in fluorescence microscope. The slides were examined using Leica TCS SPE confocal microscope (Leica Microsystems, Wetzlar, Germany).

**Bone marrow-derived macrophages (BMDMs) and infection.** Isolated femurs and tibia were flushed with incomplete RPMI, and the precursor cells were cultured in RPMI supplemented with 30% L929 cell-conditioned medium and 20% FBS. After 7 days in culture, the mature BMDMs were harvested and infected with stationary-phase or metacyclics promastigotes at different MOI, depending on the experiments performed (indicated in figure legends). In killing experiments, free parasites were washed, and fresh media was added to the infected cultures after 1 h of infection. The leishmanicidal activity of the cells was determined at different times post-infection by counting the *Giemsa*-stained cytospin preparations under a light microscope with a ×100 objective. Infection rate was determined by scoring the % of infected macrophages (100 BMDMs counted), the number of intracellular amastigotes per infected BMDM, and the frequency of amastigotes per infected cell.

**Extracellular vesicles purification and assays.** Parasites were washed with PBS, resuspended in RPMI 1640 medium without FBS and phenol red, and incubated for 4 h at 37 °C, as previously described[55]. Then, parasites were separated from culture supernatants by centrifugation at 3000 rpm for 10 min at 25 °C, and then the supernatant was centrifuged at 10,000 rpm for 15 min to clear the debris. Pellets were removed and the resulting supernatants concentrated and EVs obtained as previously described[39]. EVs were analyzed and characterized by nanoparticle tracking using NanoSight NS300 system (Malvern instruments, Malvern, UK) equipped with 405 nm laser. Vesicles were collected and analyzed using NTA software (version 3.2.16). For BMDMs stimulation, $10^7$ purified EVs from both clones were added into the 24-well plates ($5 \times 10^5$ macrophages/well) at the moment of the infection. For LRV transfer assays, $3 \times 10^7$ parasites in stationary-phase were incubated with *L.g.*+-derived EVs for 2 h in Schneider's medium, as previously described[39]. After extensive washing (at least five times with PBS 1×),

parasites were isolated for RNA extraction and PCR analysis, or to perform infections (ELISA and Killing assays) in BMDMs.

**ELISA assay.** IL-1β and TNF-α production was assessed using IL-1β and TNF-α (BD Biosciences) ELISA KIT. In vitro IL-1β production was analyzed in the cell-free supernatants harvested from the BMDMs pre-stimulated with 500 ng/ml of ultrapure LPS (InvivoGen), 10 ng/ml of TNF-α (eBioscience) or 300 ng/mL of PAM(3)CSK(4) (InvivoGen) for 4 h and subsequently infected with stationary phase or metacyclics promastigotes at different MOI.

**Endogenous caspase-1 staining using FAMYVADFMK.** BMDMs were cultured and infected with stationary phase *L.g.*− or *L.g.*+ at MOI 10. After 24 h of infection, BMDMs were stained for 1 h with FAM–YVAD–fluoromethyl ketone (FAM–YVAD–FMK; Immunochemistry Technologies), as recommended by the manufacturer's instructions. The active Casp1 was then measured by flow cytometry. The data were acquired on a FACS ACCURI C6 flow cytometer (BD Biosciences) and analyzed with the FlowJo software (Tree Star).

**Western blotting.** A total of $5 \times 10^5$ BMDMs were seeded per well and then infected with *L.g.*− or *L.g.*+ for 9 or 24 h. The supernatants were discarded and cells were lysed in a buffer containing 100 mM NaCl, 20 mM Tris (pH 7.6), 10 mM EDTA (pH 8), 0.5% SDS, and 1% Triton X-100 with protease inhibitor mixture (Roche) and incubated 20 min on ice. Sample buffer was added to lysates and samples were boiled for 5 min. Proteins were then separated by SDS-PAGE in 15% gel and transferred to nitrocellulose membranes. The following primary antibodies were used: anti-NLRP3 (1:1000; Adipogen; catalog number AG-20B-0014), anti-ASC (1:1000; Santa Cruz; catalog number sc-33958) anti-Atg5 (1:1000; Abcam; catalog number 108327), anti-LC3 (1:1000; Sigma; catalog number L7543), and anti β-actin (1:1000; Santa Cruz; catalog number sc-47778). Specific secondary antibodies were used afterward. In Human macrophages, LC3-II quantification was normalized to β-actin values using ImageJ software. Images of uncropped and unprocessed western blottings are provided in the Source Data file.

**Immunoblotting for Casp1 p20 and IL-1β p17.** A total of $10^7$ BMDMs were seeded per well, primed with ultrapure LPS (500 ng/ml) for 4 h and then infected with *L.g.*− or *L.g.*+ for 48 h. The supernatants were collected and precipitated with 50% trichloroacetic acid (TCA) and acetone. After their clarification by centrifugation, the cells were lysed in RIPA buffer (10 mM Tris-HCl, pH 7.4, 1 mM EDTA, 150 mM NaCl, 1% Nonidet P-40, 1% deoxycholate, and 0.1% SDS) containing protease inhibitor cocktail (Roche). The lysates and supernatants were resuspended in Laemmli buffer, boiled, resolved by SDS-PAGE and transferred (Semidry Transfer Cell, Bio-Rad) to a nitrocellulose membrane (GE Healthcare). The membranes were blocked in Tris-buffered saline (TBS) with 0.01% Tween-20 and 5% non-fat dry milk. The rat anti-Casp1 p20 monoclonal antibody clone 4B4 (Genentech) (1:500), goat anti-IL-1β p17 subunit (Sigma-Aldrich, 1:250) and specific anti-rat or anti-goat horseradish peroxidase-conjugated antibodies (1:3000; KPL) were diluted in blocking buffer for the incubations. The ECL luminol reagent (GE Healthcare) was used for the antibody detection.

**ROS detection.** In order to detect intracellular ROS production, BMDMs were incubated with stationary-phase promastigotes of *L.g.*− or *L.g.*+ at an MOI of 10 or stimulated with Rotenone (50 μM) or PMA (500 ng/ml) for 90 min. Next, pre-warmed H2DCFDA (10 μM) and MitoSOX Red dye (2.5 μM) were added to the cells for 30 min at 37 °C. The cells were harvested and immediately analyzed by flow cytometry (FACS).

**K+efflux assay.** To detect the ammount of potassium (K+) efflux upon *L.g.* infection, BMDMs were infected with both clones and incubated with the Asante Potassium Green-2 dye (APG-2, TEFLabs, Austin, TX, USA), that stains K+ specifically. Cells were then analyzed in a High Content Confocal equipment to determine potassium intracellular levels.

**qPCR for inflammatory genes.** A total of $10^6$ BMDMs were infected with both clones of *L.g.* or stimulated with Poly:IC (5 μg/mL) as a positive control, for 6 or 24 h. Cells were washed with phosphate-buffered saline, and resuspended and lysed in 500 μL of Trizol (Life Technologies). Total RNA was isolated according to the manufacturer's instructions. The complementary DNA (cDNA) was prepared from 1 μg of total RNA with random hexamers using the kit *High-Capacity cDNA Reverse Transcription Kit* (Applied Biosystems) according to the manufacturer's instructions. The PCR was performed by 40 cycles, using gene specific primers (*Nlrp3* F: 5′-GTGGTGACCCTCTGTGAGGT-3′–*Nlrp3* R: 5′-TCTTCCTGGAG CGCTTCTAA-3′); (*Asc* F: 5′-CCAGTGTCCCTGCTCAGAGT-3′–*Asc* R: 5′-TCA TCTTGTCTTGGCTGGTG-3′); (*Casp1* F: 5′-AGATGCCCACTGCTGATAGG-3′– *Casp1* R: 5′-TTGGCACGATTCTCAGCATA-3′); (*Il1b* F: 5′-CCAAGCAACGACA AAATACC-3′–*Il1b* R: 5′-GTTGAAGACAAACCGTTTTTCC-3′); *Tnfa* F: 5′-TGT GCTCAGAGCTTTCAACAA-3′–*Tnfa* R: 5′-CTTGATGGTGGTGCATGAGA-3′); *Ifnb* F: 5′-TCCGAGCAGAGATCTTCAGGA-3′–*Ifnb* R: 5′-TGCAACCACCACTC ATTCTGAG-3′); *Il12* F: 5′-TGGTTTGCCATCGTTTTGCTG-3′–*Il12* R: 5′-AC

AGGTGAGGTTCACTGTTTCT-3′). RT-PCR products were analyzed on a 1.5% agarose gel in 1X TAE buffer. The real-time quantitative reaction was performed in the Viia 7 Real-Time PCR System (Applied Biosystems). Results were analyzed using the $2^{-\Delta\Delta CT}$ method.

**BMDMs transduction with lentivirus encoding LC3-GFP.** HEK293 cells were transfected with the lentiviral vector pHR SINCSGWNotI-GFP-LC3 and the helper plasmids pCMV8.91and pMDG (42), and recombinant viral particles were obtained. Lentivirus-containing supernatant was added to bone marrow cells previously incubated for 3 days with LCCM, followed by an overnight incubation at 37 °C in 5% $CO_2$. The next day, the supernatant was replaced with fresh lentiviral supernatant followed by overnight incubation at 37 °C in 5% $CO_2$. Then, the supernatant was replaced with fresh media and cells were incubated for an additional 48 h before being used in autophagy assays.

**GFP-LC3 dots assay.** Images of GFP-LC3-transduced BMDMs were acquired by fluorescence microscopy (Leica DMI 4000B; Leica Microsystems, Heidelberg, Germany) and autophagy induction was determined by counting the number of cytoplasmic autophagosomes (GFP-LC3 dots) in non-infected and infected cells, which was possible given that the parasite's nucleus also stains with DAPI (Sigma). Each condition was assayed in triplicate, and at least 100 cells per well were evaluated.

***Atg5* silencing by shRNA.** Lentiviral particles were generated by transfecting HEK293T cells with the helper plasmids pMD2.G and psPAX2, in addition to plasmids codifying specific sequences for Atg5 or control (scramble) shRNA. The sequences used were 5-CCGGGCATCTGAGCTACCCAGATAACTCGAGTTAT CTGGGTAGCTCAGAGCTTTTTG-3 (Atg51) and 5-CCGGGGCCAAGTATCTGT CTATGATACTCGAGTATCATAGACAGATACTTGGCTTTTTG-3 (Atg5 2). The transduction of BMDMs was performed as described above. Expression of the Atg5 protein was systematically evaluated by immunoblotting.

**Statistical analysis.** For the comparison of multiple groups, two-way analysis of variance (ANOVA), followed by the Bonferroni post-test were used. The differences in the values obtained for two different groups were determined using Student's $t$ test. All the analyses were performed using the Prism 5.0 software (GraphPad, San Diego, CA). A difference was considered statistically significant when $P \leq 0.05$, and represented by *.

## Data availability

All data supporting the findings of this study are available from the corresponding author upon reasonable request. The source data underlying the figures of this manuscript are provided as a Source Data file.

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

## Acknowledgements

We would like to thank Maira Nakamura, Victória Santos and Elizabete Milani for their technical support. We are grateful to Vishva Dixit (Genentech) for providing the *Nlrp3*$^{-/-}$ and the anti-Casp1 p20 antibody and to Noboru Mizushima (University of Tokyo) for providing the *Atg5*$^{Flox/Flox}$ mice. We also thank Dr. Eurico Arruda Neto (USP) for discussions about the biology of LRV; Dr. Elisa Cupolillo and Dr. Renato P. de Almeida (FIOCRUZ-RJ) for providing the clinical isolates from CLIOC. This work was supported by grants from PEW, Training in Tropical Diseases/World Health Organization (TDR/WHO), INCTV/CNPq, and FAPESP (grants 2013/08216-2 and 2014/04684-4). R.V.H.C., A.L.N.S., and D.S.L.J. are supported by fellowships from FAPESP. M.V.G.S. and M.M.S. are supported by postdoctoral fellowships from CNPq/CAPES. D.S.Z. and A.K.C. are Research Fellows from CNPq.

## Author contributions

R.V.H.C., D.S.L.J., M.V.G.S., M.D., T.S.R., and A.L.N.S. designed and performed experiments. R.V.H.C, D.S.L.J., and D.S.Z. analyzed data, discussed hypotheses, generated figures, and wrote the paper. M.V.G.S., C.V.H., P.F.S., F.G.F., L.B.L., M.M.S., F.B.R. A., L.M.C., R.G.M.F., and A.K.C. generated tools for proper work development, discussed hypotheses, and revised the paper.

## Competing interests

The authors declare no competing interests.
