## [Peer Review File · Nature Communications]

Reviewers' comments:

Reviewer #1, expert on Leishmania-macrophage interactions (Remarks to the Author):

Leishmaniasis is caused by infection with Leishmania and disease severity correlates with infection by various species ranging from cutaneous to mucocutaneous to visceral. Several previous studies have identified the presence of a RNA virus, referred to LVR, in certain Leishmania Viannia species including *L. braziliensis* and *L. guyanensis*. The presence of LVR shows varying degrees of association with disease severity, mucocutaneous or cutaneous, with a trend of LVR1 more common in mucocutaneous than in cutaneous leishmaniasis. However, it is clear that the presence of LVR1 does not confer 100% disease severity. Results in this manuscript state that 33% of patients with mucocutaneous leishmaniasis, isolated parasites are LRV-. Thus is LRV just randomly distributed in various Leishmania species?

The authors report that the severity of leishmaniasis, comparing mucocutaneous to cutaneous caused by infection with either *Leishmania braziliensis* or *L. guyanensis*, correlates with reduced inflammasome activation and perhaps the presence of LVR.

Major concerns

The authors are using a mix of data from human lesions and data from experimental mouse models. This complicates conclusions relative to human disease as it has been extensively shown that human and mouse cells may behave differently to Leishmania infection in a number of ways. Many of the in vitro experiments could have been performed using human peripheral blood derived macrophages rather than mouse bone marrow derived macrophages. The use of mouse cells must be justified in order to correlated results from mouse studies to studies in humans. Certainly the results from the mouse studies that support the major conclusions should be validated using human cells.

From a stock culture of *L. guyanensis* LVR+ promastigotes, a single clone was selected that typed as *L. guyanensis* LVR-. Both cell lines were then used for a variety of experiments (Fig 2) Using various mouse strains and cells. Therefore, these comparisons were between a non-clonal cell line (LRV+) and a clonal cell line derived from a single clone (LVR-). Clonal lines of Leishmania are notorious for showing different behaviors in experiment animals and/or cells. Due to selection and perhaps even acquired mutations for survival in tissue culture during cloning. What is the evidence that the reported differences were not due to differences in clonal cell lines. With knock out mutants the best way to show phenotypic differences is to complement the mutant with the wild type gene (add back). This may not be possible in the current experimental design as the LRV may not have the ability to infect Leishmania cells

Comments:

Fig 1 : what is the justification of showing the Casp 1 or TNF data when the differences between CL versus MCL lesions are not statically different. It is misleading to state the averages were different when through statistics there were no significant differences. Conclusions can not be made form "trends".

Iine 132: "67% of the MCL patients were ...(LRV+). By contrast, only 32% of the CL patients were infected with LRV+ parasites"

What is the justification of highlighting only 32% of CL were LRV+ as these could over time turn into MCL. Is it not more relevant that 33% of the MCL were LRV- ?
this shows lack of correlation of LRV to disease severity and should be discussed.

Line 140: It is not clear what are the differences between Fig 1C versus 1G ? Are these different groups of patients or is 1G representative of CL patients that then developed MCL, if so are these the same patients in 1C ?

Fig 2 H: what is the justification for expressing these results as parasites per cell when the numbers are <1. How can there be 0.4 parasite per cell? Why are the results not presented as parasites per infected-cell?

Fig 3: Why were the mouse macrophages first treated with LPS and then infected with Leishmania. This is not a model of natural infection by promastigotes where naïve, not activated, macrophages are infected with promastigotes. For continuous in vivo infection of naïve or activated macrophages, it is the amastigote stage of Leishmania that infects such cells.

Line250, Fig 4D: "reduced inflammasome activation was abolished in the absence of Tlr3-/-." Although the difference in the magnitude of IL-1 production was reduced, in Tlr3-/- cells the IL 1 levels were around 25% higher than in wild type stimulated with LRV+. Please explain.

Line 379: change specie to species

Figure 8 line 1060. The authors state the data is representative of 3 experiments. Therefore the data from all three experiments should be combined into a single table or figure with ANOVA and test of means applied to test for significance.

Discussion

The authors throughout the discussion section state: "Now we demonstrate an inverse correlation between the severity of Leishmaniasis and inflammasome activation" (line 433) What data in the manuscript supports inflammasome activation as data using human cells is only shown in Figure1 where the only significant results of difference in IL-1 or Casp-1 levels, levels of TNF were not significantly different. Is IL-1 and Casp-1 sufficient to "demonstrate" inflammasome activation and/or whether the human cells underwent autophagy? Further experimental data is required to support such a strong statement.

Similarly "NLRP3 activation" in murine model (line 427) is stated in the same sentence as reference to activation of inflammasome in human cells, which infers is mediated by NLRP3. Where is the data using human cells showing NLRP3 activation? Authors should be more careful in using data from murine models to explain similar mechanisms in human cells, without the appropriate supporting data from human cells.

Reviewer #2, expert in Leishmania pathogenesis (Remarks to the Author):

The manuscript by de Carvalho et al. investigates innate immune pathways activated by Leishmania guyanensis harbouring or not LRV, in macrophages. This study demonstrates that LRV inhibits inflammasome activation by promoting autophagy-induced degradation of NLRP3 and ASC. This pathway is initiated by the activation of TLR3/TRIF by LRV, followed by the production of IFN- β , which ultimately induces autophagy. Inflammasome inhibition results in enhanced growth/survival of LRV+ compared to LRV- *L. guyanensis* and in heightened expression of pro-inflammatory cytokines by macrophages. The authors propose that this pathway could be responsible for initiating the devastating immunopathology observed in patients affected by mucocutaneous leishmaniasis. This work is very interesting and reveals a novel inflammasome regulatory loop initiated by a Leishmania RNA virus and involving TLR3, IFN- β , and autophagy. The study is well conducted and

controlled, and proposes a different theory for explaining the development of mucocutaneous leishmaniasis in patients infected with LRV+ *L. guyanensis*.

Major points

- 1) Based on the results, LRV+ *L.g.* grows and/or survives better in macrophages than LRV- parasites. This could affect macrophage viability. Hence, it is possible that the observed phenomena are in part a consequence of reduced host cell's survival, especially for experiments that last 48h or more. It is important to monitor the viability of cells (infected or bystander).
- 2) It is not clear why metacyclic promastigotes were not used in all experiments.
- 3) Figure 3 A and C: it is interesting to note that metacyclic promastigotes seem to induce much lower levels of IL-1 β . Would this suggest that parasites with higher infectivity, such as metacyclics and LRV+, inhibit the inflammasome? Or is it because of the slightly different LPG structure of stationary phase and metacyclic promastigotes? Do metacyclics also activate TLR3?
- 4) Very little detail is provided in the materials and methods section concerning macrophage infection. Were infections with LRV- and + *L.g.* synchronized? Were parasites that were not internalized removed from the well or were they left in culture?
- 5) Fig. 4 F and G: there seems to be a strong variability in the production of IL-1 β between experiments in 4F and 4G (e.g. about 700 pg/ml vs 130pg/ml). Why?
- 6) Fig. 5A: is the increase in LC3 puncta mainly in infected or in bystander macrophages? IFN- β could induce autophagy in bystander, uninfected cells as well, which could have additional implications in terms of disease pathogenesis. It would be better to analyse LC3 puncta in infected and uninfected cells separately.
- 7) Fig. 5 C: LC3-II is fairly high in the NI condition, compared to LC3-I. This is not the case in Fig. 5D. Why?
- 8) Fig. 7 C and G: although the differences in parasite burden between groups are clear, there is a large variation between experiments in the number of parasites present in the mouse ears, which is not the case for the Lnn and also not the case for the ear thickness: Log₁₀ 1 (which seems rather low) vs 4. Is this because *Atg5* flox/wt mice have reduced autophagy? Why are Lnn not affected?

Minor points

y-axis labeling is missing for all FAM-YVAD histograms.

The language of the paragraph describing Fig. 8 needs to be revised.

IL-1 has been shown to induce immunopathology in models of mucocutaneous leishmaniasis. Although this study does not necessarily contradict the current literature, this point should be discussed, especially if the described pathway is proposed to be responsible for enhancing disease pathogenesis and immunopathology.

Reviewer #3, expert in innate immunity (Remarks to the Author):

de Carvalho et al. examine the mechanism by which LRV acts as a virulence factor associated with the severity of mucocutaneous leishmaniasis. They demonstrate that the presence of LRV infection results in TLR3-driven autophagy that mitigates NLRP3 inflammasome activation. This is conceptually innovative and of importance in understanding leishmania pathogenesis.

However, I do have some concerns with the authors' interpretation of their findings.

Major concerns:

1. The authors suggest that autophagy-mediated degradation of NLRP3 and ASC are responsible for the limited inflammasome activation. However, previous studies have shown that the autophagic removal of damaged mitochondria (mitophagy) is responsible for diminished inflammasome activation (PMID:21151103).
2. Figure 1A and B – given that there is likely to be different levels of inflammation and edema in

cutaneous vs mucocutaneous lesions the amount of IL-1beta and caspase-1 activation should be normalized to the number of macrophages in the lesion.

3. Figure 1D-F – the CL and MCL groups should be analyzed separately rather than being pooled together.

4. Figure 4B and D – it looks like the amount of inflammasome activation by Lg- is reduced in the absence of TLR3 rather than increased inflammasome activation of the Lg+ strain. The authors should comment on this.

5. Figure 8A and B – it would be better to analyze just the *L. braziliensis* strains as combining the *L. guyanensis* strain adds an additional variable.

Minor concerns:

1. The title of the paper is confusing. It is unclear what “bursts” means. This should be reworded.

2. The abstract and title suggest that TLR3 driven type I IFN production drives autophagy. However, the authors do not provide data that shows a causative link between type I IFN and autophagy. This should be reworded.

3. In the introduction the authors state that leishmania kills millions of people worldwide. Although this is technically correct, the authors should be more specific and states deaths per year (approx.. 70,000 deaths per year).

4. There needs to be information provided in the methods section on how the patient samples were collected and processed. It is unclear to me what “cervical brushes” mean. Also information about IRB approval for human studies needs to be provided.

5. Number of mice per group for Fig 2 should be provided.

6. Figure 3G – why is the pro-IL-1beta blot appear reversed (negative)? Should a lower exposure be provided instead?

7. The vast majority of the figures are presented as one representative figure of three independent experiments. As such the statistical analysis is of technical, and not biological, replicates. Hence, they are only really telling us about the pipetting skills of the experimenter. Ideally, the experiments should be pooled together and presented as the mean +/- SEM with the appropriate statistical analysis.

8. A number of the references are not listed correctly (16, 54, 57, 61, 63).

We thank the reviewers for their critical and insightful review of our manuscript. Below and in the revised manuscript, we address all the issues raised by the reviewers, point-by-point. We performed many new experiments, expanding our original findings. We feel that in addressing the points raised by peer review, we have significantly enhanced the impact and clarity of our manuscript to allow publication in *Nature Communications*.

Reviewers' comments:

Reviewer #1, expert on Leishmania-macrophage interactions (Remarks to the Author):

Leishmaniasis is caused by infection with Leishmania and disease severity correlates with infection by various species ranging from cutaneous to mucocutaneous to visceral. Several previous studies have identified the presence of a RNA virus, referred to LRV, in certain Leishmania Viannia species including L. braziliensis and L. guyanensis. The presence of LRV shows varying degrees of association with disease severity, mucocutaneous or cutaneous, with a trend of LRV1 more common in mucocutaneous than in cutaneous leishmaniasis. However, it is clear that the presence of LRV1 does not confer 100% disease severity. Results in this manuscript state that 33% of patients with mucocutaneous leishmaniasis, isolated parasites are LRV-. Thus is LRV just randomly distributed in various Leishmania species? The authors report that the severity of leishmaniasis, comparing mucocutaneous to cutaneous caused by infection with either Leishmania braziliensis or L. guyanensis, correlates with reduced inflammasome activation and perhaps the presence of LRV.

Authors response: We thank the reviewer for raising this question. It is still unclear why some parasite species harbor LRV, while others don't (Hartley et al, Front Cell Infect Microbiol, 2012). We hypothesize that host parasite enzymatic machinery is needed for LRV to efficiently establish endosymbiosis. We speculate that *L. viannia* species, mainly *L. braziliensis*, *L. guyanensis* and *L. panamensis* might display molecules that sustain LRV replication within the parasite's kinetoplast, while parasites belonging to *Leishmania* subgenus (*L. infantum*, *L. major*) should not. This hypothesis is in accordance with a previous study showing that LRV RNA can only be transiently introduced to a LRV- strain of parasite (Armstrong et al, PNAS, 1993). We extensively tried to add back LRV to the LRV- *L. guyanensis* strain used in this study, but we never obtained any success. However, we were able to recapitulate the LRV+ phenotype by adding IFN- β or Poly:IC during infections with LRV- *L. guyanensis* (**Fig. 4g, h**). As indicated by the reviewer, it is important to mention that LRV is not the only risk factor for the development of mucocutaneous disease. These data support the hypothesis that additional factors affect the outcome of Leishmaniasis, such as the host immune response and individual differences displayed by each strain of parasite. We have clarified these issues in the Discussion section.

Major concerns

The authors are using a mix of data from human lesions and data from experimental mouse models. This complicates conclusions relative to human disease as it has been extensively shown that human and mouse cells may behave differently to Leishmania infection in a number of ways. Many of the in vitro experiments could have been performed using human peripheral blood derived macrophages rather than mouse bone marrow derived macrophages. The use of mouse cells must be justified in order to correlate results from mouse studies to studies in humans. Certainly the results from the mouse studies that support the major conclusions should be validated using human cells.

Authors response: We decided to use a mouse model of Leishmania infection, because it provides powerful genetic tools to better address the mechanisms by which LRV favors parasite replication, combining both in vitro and in vivo approaches. As suggested by the reviewer, we performed new experiments to validate our main findings in human macrophages derived from human CD14+ purified monocytes from different donors. Using human cells, we found that LRV limits inflammasome activation (shown by secretion of IL-1 β and casp-1 p20) while increasing autophagy induction by *L.g.* Mechanistically, inflammasome activation by *L.g.* is completely dependent on potassium efflux (as shown by KCl treatment, but not NaCl). Moreover, addition of Poly:IC completely rescued LRV effects in IL-1 β and casp1 p20 production by *L.g.*-. Finally, our new data demonstrate that LRV presence favors parasite persistence in human macrophages, and inhibition of the NLRP3 inflammasome via KCl or addition of Poly:IC rescued *L.g.*- capacity to survive macrophage killing. Taken together, these data validate our murine findings in human macrophages and provide a mechanism triggered by LRV to increase parasite persistence that seems to operate both in human and mouse cells. These results are now shown as **Fig. 8A-J**.

From a stock culture of L. guyanensis LVR+ promastigotes, a single clone was selected that typed as L. guyanensis LVR-. Both cell lines were then used for a variety of experiments (Fig 2) Using various mouse strains and cells. Therefore, these comparisons were between a non-clonal cell line (LRV+) and a clonal cell line derived from a single clone (LVR-). Clonal lines of Leishmania are notorious for showing different behaviors in experiment animals and/or cells.

Due to selection and perhaps even acquired mutations for survival in tissue culture during cloning. What is the evidence that the reported differences were not due to differences in clonal cell lines. With knock out mutants the best way to show phenotypic differences is to complement the mutant with the wild type gene (add back). This may not be possible in the current experimental design as the LRV may not have the ability to infect Leishmania cells

Authors response: We thank the reviewer for raising this question. As discussed above in the reviewer's first question, we tried to add back LRV to *L.g.*-, unsuccessfully. To experimentally address whether other variations, rather than LRV presence, could be responsible for the differences reported in

this study, we generated five new clonal cell lines from M4147 (*L.g.*+), and they are LRV+, as shown by PCR (**Fig. 3L**). G6PD was used as a control for an endogenous *Leishmania* gene. We then performed new experiments comparing these new 5 clones with clone 40 (*L.g.*-). By infecting wild-type BMDMs with different LRV+ clonal cells lines, we now show that *L.g.*+ and all LRV+ *L.g.* clones induce less inflammasome activation when compared to *L.g.*-. These results demonstrate that, despite any differences between clonal and non-clonal parasites, all LRV+ *L.g.* tested induce less inflammasome activation compared to *L.g.*-. Moreover, the fact that LRV limits IL-1 β production via TLR3 (as shown by the experiments with Poly:IC and using *Tlr3*^{-/-} macrophages) also argues in favor of the direct effect of LRV dsRNA in inflammasome blockage. Importantly, the data obtained with different *L.b.* clinical isolates demonstrate that, regardless of several variations between the parasites used, LRV+ isolates induce less inflammasome activation than LRV- parasites.

Comments:

Fig 1 : what is the justification of showing the Casp 1 or TNF data when the differences between CL versus MCL lesions are not statically different. It is misleading to state the averages were different when through statistics there were no significant differences. Conclusions can not be made form "trends".

Authors response: We apologize for the mistake and corrected the text in the results section, mentioning that the levels of Casp1 (CL/MCL patients) and TNF- α (LRV+/LRV- patients) are not different.

line 132: "67% of the MCL patients were ...(LRV+). By contrast, only 32% of the CL patients were infected with LRV+ parasites"

What is the justification of highlighting only 32% of CL were LRV+ as these could over time turn into MCL. Is it not more relevant that 33% of the MCL were LRV- ?

this shows lack of correlation of LRV to disease severity and should be discussed.

Authors response: We have adjusted the text accordingly. Based on the patient's data from this study, we show that LRV presence increases 3 times the likelihood of developing MCL (**Fig. 1G**), which together with previous studies, suggests that the LRV is an important risk factor associated with disease severity. However, LRV is not the only risk factor for MCL. These factors help to explain why certain patients develop MCL in the absence of LRV and why some patients never turn into MCL even in the presence of LRV. As suggested by the reviewer, we have added these comments in the Discussion section.

Line 140: It is not clear what are the differences between Fig 1C versus 1G ? Are these different groups of patients or is 1G representative of CL patients that then developed MCL, if so are these the same patients in 1C?

Authors response: Fig. 1C and Fig. 1G were plotted within the same group of

patients (n=49), but in different ways. Fig 1C show the frequency of LRV+/LRV- patients plotting by the clinical forms of the disease. Fig. 1G show the same patients plotted by the frequency of CL and MCL in LRV- and LRV+ patients. This is now clarified in the Results section.

Fig 2 H: what is the justification for expressing these results as parasites per cell when the numbers are <1. How can there be 0.4 parasite per cell? Why are the results not presented as parasites per infected-cell?

Authors response: As suggested by the reviewer, we reanalyzed our data and plotted the data as “parasites per infected cell” instead of “parasites per cell”. This is now shown in Figs **2H, 4J, 4L, 6I, 8H, 8J** and **S5B**. Importantly, in all experiments showing the “average number of parasites per cell”, there is a graph from the same experiment showing the “percentage of infected cells”.

Fig 3: Why were the mouse macrophages first treated with LPS and then infected with Leishmania. This is not a model of natural infection by promastigotes where naïve, not activated, macrophages are infected with promastigotes. For continuous in vivo infection of naïve or activated macrophages, it is the amastigote stage of Leishmania that infects such cells.

Authors response: For experiments involving IL-1 β measurements and caspase-1 cleavage, we primed the BMDMs to induce transcriptional regulation of pro-IL-1 β and pro-caspase-11, which are produced in very low levels by naïve BMDMs. This phenomenon was recently reported by our group (de Carvalho et al, J. Leukoc. Biol., 2019). For additional experiments measuring inflammasome assembly and activation, priming was not necessary. This is evident in our assays showing inflammasome activation by FLICA (**Fig. 3G-L, 4A-B, 6C-D**). To clarify this, we performed infections with both clones in unprimed versus primed-BMDMs (now shown as **Fig S2A**) and also using human macrophages (**Fig. 8A and 8B**). Importantly, regardless of the use of priming for certain in vitro experiments, during in vivo infections exogenous priming is dispensable because cytokines such as TNF- α and INF- γ (present in infections in vivo) can promote transcriptional upregulation of inflammasome components.

Line250, Fig 4D: “reduced inflammasome activation was abolished in the absence of Tlr3-/-.” Although the difference in the magnitude of IL-1 production was reduced, in Tlr3-/- cells the IL 1 levels were around 25% higher than in wild type stimulated with LRV+. Please explain.

Authors response: When we used LRV+ it is expected to see a higher IL-1 β production in the *Tlr3*^{-/-} than in C57BL/6 macrophages. This occurs because the TLR3/TRIF/IFN- β /Autophagy pathway inhibits inflammasome-mediated IL-1 β production. This is shown in **Fig. 4D** and also evident in a new experiment performed with metacyclics promastigotes (**Fig. 4E**). The important point here is the demonstration that *L.g.*+ induces much less IL-1 β release compared to *L.g.*- (looking at the B6 bars), while in *Tlr3*^{-/-} and *Nlrp3*^{-/-} cells this difference no longer exists. The combined data with both stationary-phase and metacyclics

promastigotes demonstrate that LRV requires TLR3 to reduce inflammasome activation by *L.g.* and are corroborated by addition of Poly:IC, which completely rescues LRV effects in IL-1 β release during infection by *L.g.*-.

Line 379: change specie to species

Authors response: We have changed the text accordingly.

Figure 8 line 1060. The authors state the data is representative of 3 experiments. Therefore the data from all three experiments should be combined into a single table or figure with ANOVA and test of means applied to test for significance.

Authors response: We thank the reviewer for the suggestion, but there are intrinsic variations among different experiments performed with *Leishmania*, possibly because of variations in the infectivity of the parasites. The differences (delta) are always reproducible, but depending of the experiments the real values varies (for example, in some experiments we see LRV+ inducing 300 pg/mL IL-1 β and LRV- inducing 600 pg/mL and in a similar experiment we see LRV+ inducing 150 pg/mL and LRV- inducing 300 pg/mL). Because of these variations, we prefer to show one single experiment representative of at least 3 experiments that obtained similar conclusions. In this revised version of the manuscript all graph bars shown indicated the values of the triplicates and we specify in figure legends whether we are showing statistical analysis of technical or biological replicates.

Discussion

The authors throughout the discussion section state: “Now we demonstrate an inverse correlation between the severity of Leishmaniasis and inflammasome activation” (line 433) What data in the manuscript supports inflammasome activation as data using human cells is only shown in Figure1 where the only significant results of difference in IL-1 or Casp-1 levels, levels of TNF were not significantly different. Is IL-1 and Casp-1 sufficient to “demonstrate” inflammasome activation and/or whether the human cells underwent autophagy? Further experimental data is required to support such a strong statement.

Authors response: This is an important point that we addressed experimentally. Regarding to the reviewer question, caspase-1 cleavage and secretion of active IL-1 β (in macrophages IL-1 β secretion requires caspase-1 activation) are gold standard and sufficient to claim that the inflammasome is activated. TNF- α production is independent of inflammasome activation and we measured as a control in clinical samples. To address the reviewer request of additional data with human macrophages, we performed new experiments using *L.g.*- and *L.g.*+ to infect human macrophages. Our new data using human macrophages (shown as **Fig 8A-J**) support our mechanistic data obtained with mouse macrophages. The inclusion of these new data strengthen our study and provides mechanistic demonstration of the role of LRV in modulating the host immune response, both in mouse and human

macrophages.

Similarly “NLRP3 activation” in murine model (line 427) is stated in the same sentence as reference to activation of inflammasome in human cells, which infers is mediated by NLRP3. Where is the data using human cells showing NLRP3 activation? Authors should be more careful in using data from murine models to explain similar mechanisms in human cells, without the appropriate supporting data from human cells.

Authors response: As mentioned before, we performed new experiments in monocyte-derived macrophages isolated from patients to validate our data using murine macrophages. By measuring IL-1 β and casp1 p20 levels in the supernatants of non-infected, *L.g.*- and *L.g.*+ infected cells, we found that *L.g.*+ induces less inflammasome activation than *L.g.*-, while increasing autophagy induction. Moreover, we treated cells with extracellular KCl prior to infection, which is widely used to block potassium efflux, a key inducer of NLRP3 activation. Our results show that KCl, but not NaCl (control), abolishes inflammasome activation by both clones, demonstrating that *L.g.* induces NLRP3-dependent IL-1 β release in human macrophages, and LRV limits this process.

Reviewer #2, expert in Leishmania pathogenesis (Remarks to the Author):

The manuscript by de Carvalho et al. investigates innate immune pathways activated by Leishmania guyanensis harbouring or not LRV, in macrophages. This study demonstrates that LRV inhibits inflammasome activation by promoting autophagy-induced degradation of NLRP3 and ASC. This pathway is initiated by the activation of TLR3/TRIF by LRV, followed by the production of IFN- β , which ultimately induces autophagy. Inflammasome inhibition results in enhanced growth/survival of LRV+ compared to LRV- L. guyanensis and in heightened expression of pro-inflammatory cytokines by macrophages. The authors propose that this pathway could be responsible for initiating the devastating immunopathology observed in patients affected by mucocutaneous leishmaniasis.

This work is very interesting and reveals a novel inflammasome regulatory loop initiated by a Leishmania RNA virus and involving TLR3, IFN- β , and autophagy. The study is well conducted and controlled, and proposes a different theory for explaining the development of mucocutaneous leishmaniasis in patients infected with LRV+ L. guyanensis.

Major points

1) Based on the results, LVR+ L.g. grows and/or survives better in macrophages than LRV- parasites. This could affect macrophage viability. Hence, it is possible that the observed phenomena are in part a consequence of reduced host cell's survival, especially for experiments that last 48h or more. It is important to monitor the viability of cells (infected or bystander).

Authors response: We thank the reviewer for raising this question. We

addressed this point experimentally by performing a FACS assay in which we pre-incubated *L.g.*- and *L.g.*+ with the green-fluorescent CFSE dye, and then infected BMDMs up to 96 hours. The results demonstrate that the percentage of infected cells by both clones are similar at 1 hour, but while *L.g.*+ survives within cells up to 96 hours, *L.g.*- is controlled over time (**Fig. 2I**). This new data are in agreement with our *Giemsa* staining experiment (**Fig. 2G-H**). Importantly, no cell death was observed in non-infected and infected cells at 1 and 24 hours of infection. We detected a slight decrease in cell viability from 48 to 96 hours with both clones (**Fig. 2I-M**). These new data demonstrate that *L.g.* infection do not induce significant cell death and the presence of LRV does not affect this process.

2) *It is not clear why metacyclic promastigotes were not used in all experiments.*

Authors response: We used stationary phase instead of metacyclics in some experiments because of the limitations to obtain high numbers of metacyclic parasites. But, importantly, the key findings of this paper (in vitro Killing, ELISAs and in vivo assays) were validated using both stationary-phase and metacyclic parasites. In this revised version of this manuscript, we included an experiment measuring inflammasome activation in *Nlrp3*^{-/-} and *Tlr3*^{-/-} BMDMs using with metacyclic promastigotes (**Fig. 4E**). The data is consistent with stationary phase parasites (**Fig. 4D**). We believe that combining stationary-phase promastigotes data, which is widely used in the literature, with the use of the highly-infective metacyclic parasites, we strengthen our conclusions about LRV effects in the host's innate immune response.

3) *Figure 3 A and C: it is interesting to note that metacyclic promastigotes seem to induce much lower levels of IL-1 β . Would this suggest that parasites with higher infectivity, such as metacyclics and LRV+, inhibit the inflammasome? Or is it because of the slightly different LPG structure of stationary phase and metacyclic promastigotes? Do metacyclics also activate TLR3?*

Authors response: In **Fig. 3A** we used stationary-phase promastigotes at an MOI 10. The previous **Fig. 3C** is an independent experiment in which we used metacyclic promastigotes at MOI of 1 and 4. Thus, it is difficult to compare two independent experiments. To address the reviewer question, we performed a complete experiment comparing stationary-phase and metacyclics promastigotes using low (MOI3) and high (MOI10) MOI. As an additional control, we also used the procyclic fraction generated after FICOLL purification. This new experiment suggests that these different forms of the parasites induce similar inflammasome activation. These new data replace the former **Fig. 3C** and are shown as **Fig. 3D-F**.

4) *Very little details are provided in the materials and methods section concerning macrophage infection. Were infections with LRV- and + *L.g.* synchronized? Were parasites that were not internalized removed from the*

well or were they left in culture?

Authors response: We thank the reviewer for this comment. We have included additional details in the materials and methods (Section “Parasite culture and infection *in vivo* and *in vitro*”) providing detailed information. We do synchronize the phases of the parasites by passaging twice both clones in log phase (2 days of growth in fresh Schneider’s medium, 10^5 parasites/mL). After that, we passage again to generate cultures for infection (also 10^5 parasites/mL of Schneider) and wait until they reach stationary-phase (usually 5 days after the passage). For experiments that we do not need the cell-free supernatants, such as *Giemsa* and FACS, we infected with both clones for 1 hour, and then wash each well twice with fresh PBS 1X, replacing with fresh RPMI 10% FBS. For ELISA, Western blotting and other inflammasome readout assays, we do not washout the parasites because there is a significant proportion of IL-1 β secretion in the initial hours of infection. This has been reported by two different studies from our group (de Carvalho et al, Cell Rep, 2019; Lima-Junior et al, J Immunol, 2017).

5) Fig. 4 F and G: there seems to be a strong variability in the production of IL-1 β between experiments in 4F and 4G (e.g. about 700 pg/ml vs 130pg/ml). Why?

Authors response: The reviewer is correct. We see a variation between inflammasome activation when we compare different *Leishmania* experiments. We believe that happens because of variations in *Leishmania* infectivity, which is affected according with cultivation and time to reach stationary phase. However, it is important to state that the differences (delta) are always reproductively between the different experiments and all the experiments shown are one experiment representative of multiple experiments performed.

6) Fig. 5A: is the increase in LC3 puncta mainly in infected or in bystander macrophages? IFN- β could induce autophagy in bystander, uninfected cells as well, which could have additional implications in terms of disease pathogenesis. It would be better to analyse LC3 puncta in infected and uninfected cells separately.

Authors response: As suggested by the reviewer, we reanalyzed the Fig. 5A experiment and plotted the average number of LC3 puncta per non-infected or infected cell. We see an increased autophagy induction in non-infected cells only after 24h stimulation (**Fig 5A**). This is consistent with our assertion that IFN- β promote autophagy and can act in non-infected cells present in the infected cultures. We appreciate the reviewer suggestion and discuss this in the manuscript.

7) Fig. 5 C: LC3-II is fairly high in the NI condition, compared to LC3-I. This is not the case in Fig. 5D. Why?

Authors response: In the autophagy field it is common to see a variable degree of basal autophagy in each experiment, probably influenced by the

time of culture and cell concentration used in the experiments. Importantly, LRV effects in autophagy induction were constantly detected in all experiments performed and we always use appropriate internal loading controls (β -actin) to draw conclusions in each individual experiment.

8) *Fig. 7 C and G: although the differences in parasite burden between groups are clear, there is a large variation between experiments in the number of parasites present in the mouse ears, which is not the case for the Lnn and also not the case for the ear thickness: Log10 1 (which seems rather low) vs 4. Is this because Atg5 flox/wt mice have reduced autophagy? Why are Lnn not affected?*

Authors response: As the reviewer mentioned, the number of *L.g.*- parasites observed in *Atg5^{F1/+}* mice are higher than in WT mice when we compare these two independent experiments. We believe it is difficult to draw solid conclusions comparing two different experiments performed at different times. We did not perform a side-by-side experiment comparing *Atg^{F1/+}* and C57BL/6 mice to test this hypothesis. As the reviewer mentioned, it is possible that the lack of only one *Atg5* allele has indeed affected parasite control in the ears. Importantly, we decided to use littermate controls *Atg^{F1/+}* mice and *Atg5^{F1/F1}* mice for this experiment because they are the best control for these conditional knockout animals. Of note, these mice lack *Atg5* only in myeloid cells (monocytes, macrophages and neutrophils), and that could be the reason for this possible effect in the ear and not in the lymph node, which is composed of lymphocytes in its majority. Regardless to these speculations, our data unequivocally demonstrate that *L.g.*+ parasites replicate better in lymph nodes and ears of *Atg5^{F1/+}* (as seen in C57BL/6 mice) and this phenomenon is abolished in *Atg5^{F1/F1}* animals.

Minor points

y-axis labeling is missing for all FAM-YVAD histograms.

Authors response: We thank the reviewer for pointing this out. We have corrected both FAM-YVAD histograms (Fig. 3 and 4) and H2DCFDA histogram (Fig. S4).

The language of the paragraph describing Fig. 8 needs to be revised.

Authors response: We have revised this.

IL-1 has been shown to induce immunopathology in models of mucocutaneous leishmaniasis. Although this study does not necessarily contradict the current literature, this point should be discussed, especially if the described pathway is proposed to be responsible for enhancing disease pathogenesis and immunopathology.

Authors response: We have discussed this in the discussion.

Reviewer #3, expert in innate immunity (Remarks to the Author):

de Carvalho et al. examine the mechanism by which LRV acts as a virulence factor associated with the severity of mucocutaneous leishmaniasis. They demonstrate that the presence of LRV infection results in TLR3-driven autophagy that mitigates NLRP3 inflammasome activation. This is conceptually innovative and of importance in understanding leishmania pathogenesis.

However, I do have some concerns with the authors interpretation of their findings.

Major concerns:

1. The authors suggest that autophagy-mediated degradation of NLRP3 and ASC are responsible for the limited inflammasome activation. However, previous studies have shown that the autophagic removal of damaged mitochondria (mitophagy) is responsible for diminished inflammasome activation (PMID:21151103).

Authors response: This is an interesting question that we addressed experimentally. To address a possible effect of LRV-induced mitophagy, we assessed the generation of mitochondrial ROS upon *L.g.*- and *L.g.*+ infection. We found that both clones induce similar levels of total ROS and they did not induced mitochondrial ROS (**Fig. S4 A,B**). To further evaluate the contribution of mitophagy in LRV-mediated inflammasome inhibition, we measured IL-1 β secretion in BMDMs lacking the E3-ubiquitin ligase *Parkin* (*Prkn*^{-/-}), which is essential in the mitophagic process. Our data suggest that *L.g.*+ induces less IL-1 β secretion than *L.g.*+ in both *Prkn*^{-/-} and *Prkn*^{+/+} BMDMs. This data is now shown as **Fig. 6F** and suggests that the modulation of inflammasome activation by LRV occurs independent on mitochondrial ROS and mitophagy.

2. Figure 1A and B – given that there is likely to be different levels of inflammation and edema in cutaneous vs mucocutaneous lesions the amount of IL-1beta and caspase-1 activation should be normalized to the number of macrophages in the lesion.

Authors response: The human samples that we obtained were cervical brushes obtained from patients lesions (More details in the “Materials and Methods” section). The material were collected from patients during clinical evaluation and stored in RNA later. Therefore, the normalization suggested by the reviewer is not possible. Importantly, we normalized the individual levels of IL-1 β , TNF- α and Caspase-1 to the respective total amount of protein in each sample, which was quantified by Bradford. As a result, our data in Figure 1 show pg (of IL-1 β , TNF- α or Casp1 p20) per mg of total protein.

3. Figure 1D-F – the CL and MCL groups should be analyzed separately rather than being pooled together.

Authors response: We thank the reviewer for the suggestion but we think it is

appropriate to plot **Fig. 1A-B** separating the CL x MCL and **Fig. 1D-E** separating LRV+ x LRV-. We believe that this is important to emphasize that regardless of the outcome of the disease, LRV is associated with decreased inflammasome activation. Furthermore, as shown in **Fig. 1A-B**, the IL-1 β production and Casp1 p20 is very low in samples from MCL patients.

4. *Figure 4B and D – it looks like the amount of inflammasome activation by Lg- is reduced in the absence of TLR3 rather than increased inflammasome activation of the Lg+ strain. The authors should comment on this.*

Authors response: When we used LRV+ it is expected to see a higher IL-1 β production in the *Tlr3*^{-/-} than in C57BL/6 macrophages. This occurs because the TLR3/TRIF/IFN- β /Autophagy pathway inhibits inflammasome-mediated IL-1 β production. This is shown in **Fig. 4B and 4D** and is also evident in a new experiment performed with metacyclics promastigotes (**Fig. 4E**). The important point here is the demonstration that *L.g.*+ induces much less IL-1 β release compared to *L.g.*- (looking at the B6 bars), while in *Tlr3*^{-/-} and *Nlrp3*^{-/-} cells this difference no longer exists. The combined data with both stationary-phase and metacyclics promastigotes demonstrate that LRV requires TLR3 to reduce inflammasome activation by *L.g.* and are corroborated by addition of Poly:IC, which completely rescues LRV effects in IL-1 β release during infection by *L.g.*-.

5. *Figure 8A and B – it would be better to analyze just the L. braziliensis strains as combining the L. guyanensis strain adds an additional variable.*

Authors response: Following the reviewer suggestion, we now plot *L. guyanensis* clones and clinical isolates in Fig. 9A, while in Fig. 9B we plot only *L. braziliensis* clinical isolates. We thank the reviewer for the suggestion.

Minor concerns:

1. *The title of the paper is confusing. It is unclear what “bursts” means. This should be reworded.*

Authors response: We have modified the title of the paper as suggested.

2. *The abstract and title suggest that TLR3 driven type I IFN production drives autophagy. However, the authors do not provide data that shows a causative link between type I IFN and autophagy. This should be reworded.*

Authors response: Our data in figure **Fig. 5D** indicates that addition of Poly:IC or IFN- β induce autophagy in non-infected cells and also rescue the *L.g.*- capacity to induce autophagy. Moreover, in **Fig. 6E** we demonstrate that addition of IFN- β during *L.g.*- infection rescues the LRV effects in inflammasome inhibition in *LysM*^{Cre/+} *Atg5*^{Fl/+} BMDMs, but not in *LysM*^{Cre/+} *Atg5*^{Fl/Fl} BMDMs. These data support a role of LRV in inducing type I IFN production, which triggers autophagy to limit inflammasome activation during *Leishmania* infection.

3. *In the introduction the authors state that leishmania kills millions of people worldwide. Although this is technically correct, the authors should be more specific and states deaths per year (approx. 70,000 deaths per year).*

Authors response: We have adjusted the text accordingly.

4. *There needs to be information provided in the methods section on how the patient samples were collected and processed. It is unclear to me what “cervical brushes” mean. Also information about IRB approval for human studies needs to be provided.*

Authors response: We thank the reviewer for the suggestion. In the Material and Methods section, we now describe in details the recruitment of patients, Ethics Statement and human sample collection and processing. Briefly, sample collection was performed using a sterile cervical brush placed in direct contact with the internal edge of the lesions. Sampling was performed for both RNA extraction and cytokine/protein quantification. The collected material was immediately stored in an RNAlater solution (Ambion, Austin, TX, USA) for preservation of molecular contents and was stored at -20°C until the time of analysis.

5. *Number of mice per group for Fig 2 should be provided.*

Authors response: This information is now provided in Figure 2 legend.

6. *Figure 3G – why is the pro-IL-1beta blot appear reversed (negative)? Should a lower exposure be provided instead?*

Authors response: We ran a new gel with the cellular extract samples from this experiment and now provide an improved Western Blotting image for pro-IL-1 β .

7. *The vast majority of the figures are presented as one representative figure of three independent experiments. As such the statistical analysis is of technical, and not biological, replicates. Hence, they are only really telling us about the pipetting skills of the experimenter. Ideally, the experiments should be pooled together and presented as the mean +/- SEM with the appropriate statistical analysis.*

Authors response: We thank the reviewer for the suggestion, but there are intrinsic variations among different experiments performed with *Leishmania*, possibly because of variations in the infectivity of the parasites. The differences (delta) are always reproducible, but depending of the experiments the real values varies (for example, in some experiments we see LRV+ inducing 300 pg/mL IL-1 β and LRV- inducing 600 pg/mL and in a similar experiment we see LRV+ inducing 150 pg/mL and LRV- inducing 300 pg/mL). Because of these variations, we prefer to show one single experiment representative of at least 3 experiments that obtained similar conclusions. In this revised version of the manuscript all graph bars shown indicated the

values of the triplicates and we specify in figure legends whether we are showing statistical analysis of technical or biological replicates.

8. *A number of the references are not listed correctly (16, 54, 57, 61, 63).*

Authors response: We have corrected that.

Reviewers' comments:

Reviewer #1 (Remarks to the Author):

The authors have addressed many of the points raised by the reviewers and presented new data Using human derived blood leukocytes to validate their results using experimental mouse strains. The authors stated that the *L. guyanensis* LRV- clone, that showed reduced pathogenesis, could not be infected with the LRV purified virus and this the phenotype could not be complemented with LRV. This would be an important control and the authors did not discuss a recent paper where it was shown that LRV together with exosome material was infective and restored the phenotype of LRV- promastigotes to a LRV+ phenotype (Nature Microbiology 4, 714–723, 2019). Therefore, the inability in the current study to complement the LRV- phenotype requires further explanation and/or experimentation.

Reviewer #2 (Remarks to the Author):

All my concerns have been adequately addressed.

Reviewer #3 (Remarks to the Author):

The authors have addressed my previous concerns.

Reviewers' comments:

Reviewer #1 (Remarks to the Author):

The authors have addressed many of the points raised by the reviewers and presented new data Using human derived blood leukocytes to validate their results using experimental mouse strains.

*The authors stated that the *L. guyanensis* LVR- clone, that showed reduced pathogenesis, could not be infected with the LRV purified virus and this the phenotype could not be complemented with LRV. This would be an important control and the authors did not discuss a recent paper where it was shown that LRV together with exosome material was infective and restored the phenotype of LRV- promastigotes to a LRV+ phenotype (Nature Microbiology 4, 714–723, 2019). Therefore, the inability in the current study to complement the LRV- phenotype requires further explanation and/or experimentation.*

Authors response: Following the reviewer's suggestion, we performed new experiments with purified extracellular vesicles as described in Atayde et al, Cell Rep, 2015. The experiment is now shown as **Figure 7**. Initially, we confirmed that LRV is found within EVs produced by *L.g.*+ strain (**Fig. 7a**), confirming previous report by Olivier's group. Both *L.g.*+ and *L.g.*- strains of *Leishmania* produce EVs and the presence of the virus does not interfere with size and production of EVs (**Fig. 7b**). Next, we treated macrophages with EVs produced by *L.g.*- or *L.g.*+ in the moment of infection and found that EVs from *L.g.*+ (but not from *L.g.*-) rescued LRV effects in inflammasome activation (**Fig. 7c,d**). In support to our data, this was dependent on TLR3 and autophagy. Finally, we incubated stationary-phase promastigotes of *L.g.*- (and *L.g.*+ as control) with EVs produced by *L.g.*+ and generated a complemented strain (*L.g.*- EVs LRV+) (**Fig. 7e**). This strain recapitulated the phenotypes of the wild type *L.g.*+ strain for inflammasome activation (**Fig. 7f**) and intracellular replication (**Fig. 7g-j**).

We believe addition of this experiment provide solid and relevant data supporting our assertion that LRV operates to inhibit inflammasome activation through TLR3 and autophagy. We thank the reviewer for this suggestion.

Reviewer #2 (Remarks to the Author):

All my concerns have been adequately addressed.

Reviewer #3 (Remarks to the Author):

The authors have addressed my previous concerns.

REVIEWERS' COMMENTS:

Reviewer #1 (Remarks to the Author):

The authors have more than adequately addressed the reviewers' comments. They should be commended for performing the extra work that adds substantially to this manuscript

The wording of the title is still award English with the use of "worsens" that is non-scientific term and not usually associated with quantitating an infection.

A more scientific term would be "exacerbates"

Reviewers' comments:

Reviewer #1 (Remarks to the Author):

The authors have more than adequately addressed the reviewers' comments. They should be commended for performing the extra work that adds substantially to this manuscript

The wording of the title is still award English with the use of "worsens" that is non-scientific term and not usually associated with quantitating an infection.

A more scientific term would be "exacerbates"

Authors response: We thank all the reviewers for their important contributions to this paper. Following the reviewer's suggestion, we modified the title of our manuscript, which is now "**Modulation of host innate immune responses by *Leishmania* RNA virus exacerbates Leishmaniasis**".